# BZLF1 interacts with chromatin remodelers promoting escape from latent infections with EBV

Marisa Schaeffner[1,2,*], Paulina Mrozek-Gorska[1,2,*], Alexander Buschle[1,2], Anne Woellmer[1,2], Takanobu Tagawa[1,2], Filippo M. Cernilogar[3], Gunnar Schotta[3,4], Nils Krietenstein[3], Corinna Lieleg[3], Philipp Korber[3], Wolfgang Hammerschmidt[1,2]

A hallmark of EBV infections is its latent phase, when all viral lytic genes are repressed. Repression results from a high nucleosome occupancy and epigenetic silencing by cellular factors such as the Polycomb repressive complex 2 (PRC2) and DNA methyltransferases that, respectively, introduce repressive histone marks and DNA methylation. The viral transcription factor BZLF1 acts as a molecular switch to induce transition from the latent to the lytic or productive phase of EBV's life cycle. It is unknown how BZLF1 can bind to the epigenetically silenced viral DNA and whether it directly reactivates the viral genome through chromatin remodeling. We addressed these fundamental questions and found that BZLF1 binds to nucleosomal DNA motifs both in vivo and in vitro. BZLF1 co-precipitates with cellular chromatin remodeler ATPases, and the knock-down of one of them, INO80, impaired lytic reactivation and virus synthesis. In Assay for Transposase-Accessible Chromatin-seq experiments, non-accessible chromatin opens up locally when BZLF1 binds to its cognate sequence motifs in viral DNA. We conclude that BZLF1 reactivates the EBV genome by directly binding to silenced chromatin and recruiting cellular chromatin-remodeling enzymes, which implement a permissive state for lytic viral transcription. BZLF1 shares this mode of action with a limited number of cellular pioneer factors, which are instrumental in transcriptional activation, differentiation, and reprogramming in all eukaryotic cells.

## Introduction

Eukaryotic DNA-binding sites are often not accessible to their cognate factors because the sites lie within epigenetically silent chromatin and are occupied by nucleosomes. Nucleosomes at binding sites constitute a physical barrier to transcription factors because their binding is often structurally incompatible with DNA wrapped around the histone octamer. Access to nucleosomal sites may be achieved through cooperative and simultaneous binding of several transcription factors that outcompete the histone octamer (Adams & Workman, 1995; Mirny, 2010). Alternatively, one class of transcription factors, termed pioneer factors (Cirillo et al, 1998, 2002; Magnani et al, 2011b; Zaret & Carroll, 2011), can bind their target sequences even on nucleosomal DNA and in silent chromatin and establish competence for gene expression through chromatin remodeling (Zaret & Mango, 2016 for a recent review). Pioneer factors either open chromatin directly through their binding or recruit chromatin modifiers and ATP-dependent chromatin-remodeling enzymes that open chromatin to allow access for the transcription machinery (Clapier & Cairns, 2009; Bartholomew, 2014; Längst & Manelyte, 2015). Such pioneer factors play key roles in hormone-dependent cancers (Jozwik & Carroll, 2012), embryonic stem cells and cell fate specification (Smale, 2010; Drouin, 2014), and cellular reprogramming (Iwafuchi-Doi & Zaret, 2014; Soufi et al, 2015). Currently, 2,000–3,000 sequence-specific DNA-binding transcription factors in human cells are known (Lander et al, 2001; Venter et al, 2001), but only about a dozen are functionally confirmed as pioneer factors.

Certain pioneer factors have peculiar structural characteristics that explain binding to nucleosomal DNA. For example, the winged-helix DNA-binding domain of the paradigm pioneer factor FoxA structurally resembles the linker histone H1, disrupts inter-nucleosomal interactions, opens chromatin, and enhances *albumin* expression in liver cells (Cirillo et al, 2002; Sekiya et al, 2009). How many other pioneer factors bind to nucleosomal DNA is less well understood, but some directly target partial DNA motifs displayed on the nucleosomal surface (Soufi et al, 2015). Subsequently, most pioneer factors recruit chromatin remodelers to their binding sites, which open silent chromatin and regulate cell-type specific gene expression (Magnani et al, 2011a; Mayran et al, 2015).

In eukaryotic nuclei, chromatin remodelers mediate the dynamics of nucleosome arrangements and participate in most DNA-dependent processes (Längst & Manelyte, 2015 for a recent

[1]Research Unit Gene Vectors, Helmholtz Zentrum München, German Research Center for Environmental Health, Munich, Germany   [2]German Center for Infection Research (DZIF), Partner Site Munich, Munich, Germany   [3]Biomedical Center, Molecular Biology, Ludwig-Maximilians-Universität Munich, Planegg, Germany   [4]Center for Integrated Protein Science Munich, Munich, Germany

Correspondence: hammerschmidt@helmholtz-muenchen.de
*Marisa Schaeffner and Paulina Mrozek-Gorska contributed equally to this work

overview). They bind to nucleosomes and convert the energy of ATP hydrolysis into the movement, restructuring, or ejection of histone octamers depending on the remodeler. Remodelers are categorized according to their ATPase subunit into four major (SWI/SNF, ISWI, INO80, and CHD) and several minor families and further differentiated by their associated subunits. This range of features reflects specialized functions found in their domains/subunits that mediate direct interactions with modified histones, histone variants, DNA structures/sequences, RNA molecules, and transcription factors. The human genome encodes 53 different remodeler ATPases (Längst & Manelyte, 2015), which are highly abundant chromatin factors with roughly one remodeling complex per 10 nucleosomes (Längst & Manelyte, 2015).

EBV infects more than 95% of the adult population worldwide with a lifelong persistence in human B cells. The key to EBV's success lies in its ingenious multipartite life cycle, which relies on different epigenetic states of viral DNA (Woellmer & Hammerschmidt, 2013). Initially, EBV establishes a latent infection in all cells it infects (Kalla et al, 2012; Hammerschmidt, 2015). Viral latency is characterized by an epigenetically silenced EBV genome that prevents the expression of all lytic viral genes but usually spares a small set of the so-called latent viral genes that remain active. Cellular factors, for example, the Polycomb repressive complex 2 (PRC2) and DNA methyltransferases, respectively, introduce repressive histone marks and 5-methyl cytosine residues into viral DNA, which ensure the repressed state of all viral lytic genes (Ramasubramanyan et al, 2012; Woellmer et al, 2012).

BZLF1 is the viral factor that acts as a molecular switch, induces the lytic, productive phase of EBV de novo synthesis, and hence abrogates transcriptional repression of viral lytic genes (Countryman & Miller, 1985; Chevallier-Greco et al, 1986; Takada et al, 1986; Countryman et al, 1987). BZLF1 binds methylated EBV DNA sequence-specifically (Bhende et al, 2004; Bergbauer et al, 2010; Kalla et al, 2012), but if and how it overcomes epigenetically repressed chromatin is not known.

BZLF1 binds to two classes of BZLF1-responsive elements (ZREs): one class contains a DNA sequence motif reminiscent of the canonical AP-1–binding site, the other class contains a sequence motif with a CpG dinucleotide, which must carry 5-methyl cytosine residues for efficient BZLF1 binding (Bhende et al, 2004; Karlsson et al, 2008; Bergbauer et al, 2010; Flower et al, 2011). Binding of BZLF1 to viral chromatin induces the loss of nucleosomes at certain but not all ZREs with higher than average nucleosome densities (see Figs 2 and 3 in Woellmer et al (2012)). The study by Woellmer et al (2012) did not determine whether the initial binding of BZLF1 and loss of nucleosomes are simultaneous events or occur sequentially nor did it identify the molecular mechanisms that underlie these events.

Here, we report that the viral factor BZLF1 acts like a pioneer transcription factor. BZLF1 binds mononucleosomal DNA in repressed lytic promoters in vivo and binds to nucleosome core particle DNA in vitro. Upon BZLF1's binding to its binding sites in silent viral chromatin their surroundings open up and become widely accessible as demonstrated in chromatin immunoprecipitation (ChIP)-seq and Assay for Transposase-Accessible Chromatin (ATAC)-seq experiments. Chromatin accessibility strictly depends on BZFL1's transactivation domain (TAD). BZLF1 interacts with two different chromatin remodelers and likely recruits them to epigenetically repressed viral chromatin. Co-precipitations identify the

transcriptional activation domain of BZLF1 as interacting with the remodeler ATPase INO80, which BZLF1 seems to tether to BZLF1-regulated viral promoters in ChIP experiments. A knock-down of INO80 reduces the activation of early lytic viral genes and virus de novo synthesis, suggesting that the BZLF1-mediated recruitment of INO80-containing remodeler complexes is an important function for viral reactivation.

# Results

## Loss of histone H3 at repressed lytic promoters follows initial lytic viral reactivation

We used our established model (Woellmer et al, 2012) to analyze the kinetics of nucleosomal loss at selected loci in EBV DNA upon lytic induction. Raji cells, a Burkitt's lymphoma–derived cell line latently infected with EBV, were engineered to contain an inducible BZLF1 allele and termed Raji p4816 cells (Fig S1). In this model, adding doxycycline triggers the expression of BZLF1 and induces viral lytic reactivation. We compared the kinetics of BZLF1's induced expression with the kinetics of nucleosome loss at selected, BZLF1-controlled lytic EBV promoters. We harvested samples from uninduced and induced Raji p4816 cells at different time points after addition of doxycycline and analyzed BZLF1 expression by Western blotting (Fig 1A). Raji p4816 cells showed a clear BZLF1 signal already 2 h post induction, and the protein level increased and reached high levels 15 h post induction. As expected, doxycycline did not induce BZLF1 in parental Raji cells (Fig 1A).

We were concerned if the level of BZLF1 protein present in induced Raji p4816 cells might exceed the levels present in EBV-positive cells that support EBV's lytic phase. To address this point, we compared the levels of BZLF1 in the B95-8 cell line with levels in our Raji cell model after induced expression of BZLF1. A small fraction of B95-8 cells spontaneously enter the lytic phase and support virus de novo synthesis (Miller et al, 1972). We found that the BZLF1 levels we reach in the Raji inducible system are in a range also found in the small fraction of B95-8 cells that undergo the lytic cycle of EBV (Buschle et al, 2019 Preprint).

Next, we performed ChIP with cross-linked viral chromatin, which had been fragmented to an average size of 200 bp and an antibody directed against histone H3 indicative of the histone octamer. We detected a partial loss of H3 at promoter sites of certain early lytic genes as reported previously (Woellmer et al, 2012), but only after 15 h post induction (Fig 1B). In contrast, H3 levels were unaffected at latent and late lytic promoters (Fig 1B) (Woellmer et al, 2012). The data indicated that BZLF1 expression clearly preceded the detectable loss of H3 at certain promoters of early lytic genes in lytically induced Raji cells.

Addition of doxycycline to mammalian cells might have adverse effects and alter transcription and chromatin structure or affect cell vitality. We tested this aspect in parental Raji cells and at doxycycline concentrations used in this and all subsequent experiments. RNA-seq experiments with induced and uninduced Raji cells did not identify any noticeable change in cellular transcription (Buschle et al, 2019 Preprint).

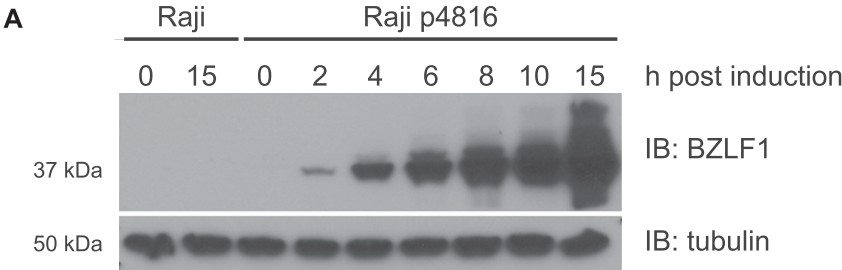

**A**

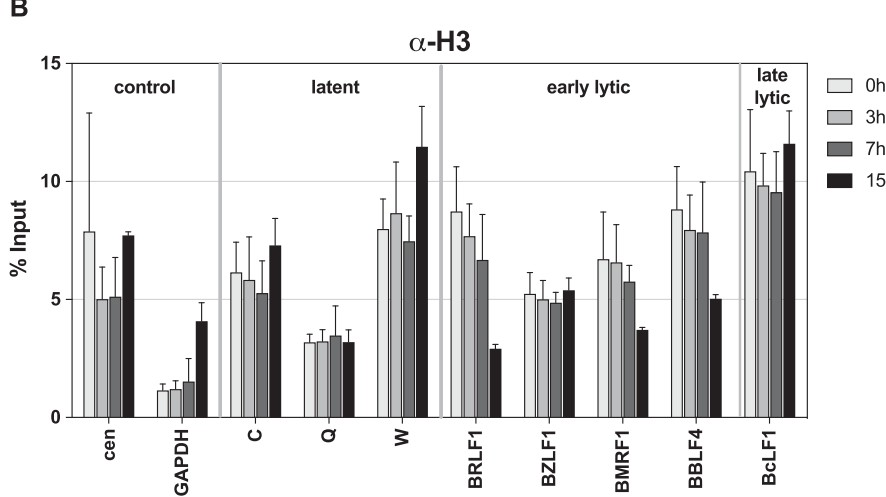

**B**

**Figure 1. BZLF1 expression precedes H3 loss at promoter sites of early lytic genes in lytically induced Raji cells.**
**(A)** The kinetics of BZLF1 protein expression in parental Raji cells and Raji p4816 cells was analyzed by immunoblotting (IB) with a BZLF1-specific antibody (top panel) at the indicated time points (hours post induction). Immunodetection of tubulin (bottom panel) served as loading control. **(B)** ChIP directed against histone H3 (#1791; Abcam) of lytically induced Raji p4816 cells at the indicated time points of induction. Mean and SD from three independent experiments are shown. Primer information can be found in Table S2.

### BZLF1 binds mononucleosomal DNA in viral lytic promoters in vivo

It is unclear if BZLF1 can bind nucleosomal DNA directly or relies on a mechanism that exposes ZRE motifs, presumably by nucleosome eviction before BZLF1's binding. The former possibility would correspond to pioneer factor–like binding of BZLF1 at nucleosomal sites; the latter would involve additional unknown factors required to facilitate BZLF1's binding. To examine the first possibility, we looked for co-occupancy of BZLF1 and histone octamers at ZREs in our model cell line in vivo. We performed ChIP and sequential ChIP (ReChIP) experiments with latent phase chromatin at different time points after lytic induction with antibodies directed against BZLF1 and the histone mark H3K4me1. Before these ChIP experiments, the chromatin had been cross-linked and sheared to mononucleosomal size by sonication and limited MNase treatment. We chose an antibody directed against the specific H3K4me1 histone mark for two reasons: (i) in our hands, antibodies directed against pan H3 (and other core histone proteins tested) performed adequately in classical ChIP experiments but poorly in ReChIP experiments. In contrast, antibodies directed against certain histone marks such as H3K4me1 were well suited for the technically challenging ReChIP experiments. (ii) Upon induction of EBV's lytic phase, the prevalence of the H3K4me1 modification increased slightly at early lytic promoters over time (Figs 2A and S2), which improves the chances to detect possible interactions of BZLF1 with ZREs embedded in nucleosomal DNA in ReChIP experiments.

BZLF1's binding (Fig 2A, left panel) was detected at most early lytic promoters within 4 h after lytic induction. A modest, up to twofold increase of H3K4me1 (right panel) became obvious 7 h post induction. Results from our three individual experiments are shown in Fig S2. ReChIP experiments were carried out with either order of the two antibodies (Fig 2B). The results demonstrated the co-occupancy of BZLF1 and histone octamers on the same DNA molecules in the promoter regions of early lytic genes 7 and 15 h post induction. ReChIP experiments in which a nonspecific IgG antibody replaced either of the two antibodies served as negative controls for the second precipitation step (Fig 2C). Carryover of chromatin complexes from the first round of ChIP experiments with antibodies directed against BZLF1 or H3K4me1 was low (Fig 2C, left panel) or negligible (Fig 2C, right panel), respectively. The results suggested that BZLF1 can bind directly to nucleosomal DNA in vivo.

### BZLF1 binds mononucleosomal DNA in vitro

We verified our in vivo finding in a defined and unambiguous in vitro system using electromobility shift assays (EMSAs) with purified BZLF1 protein and ZRE-containing DNA fragments in their free states or bound as mononucleosomes (Fig 3). The latter bound fragments serve as surrogates for viral chromatin in its repressed state.

The promoter of the early lytic gene BBLF4, which encodes the viral DNA helicase, harbors five ZREs of 9 base pairs in length. All five ZREs contain CpG dinucleotides and show a methylation-dependent binding of BZLF1 (Bergbauer et al, 2010). We prepared three 156-bp-long DNA fragments derived from this promoter region that differed in the positioning of two ZREs, ZRE 3 and ZRE 4

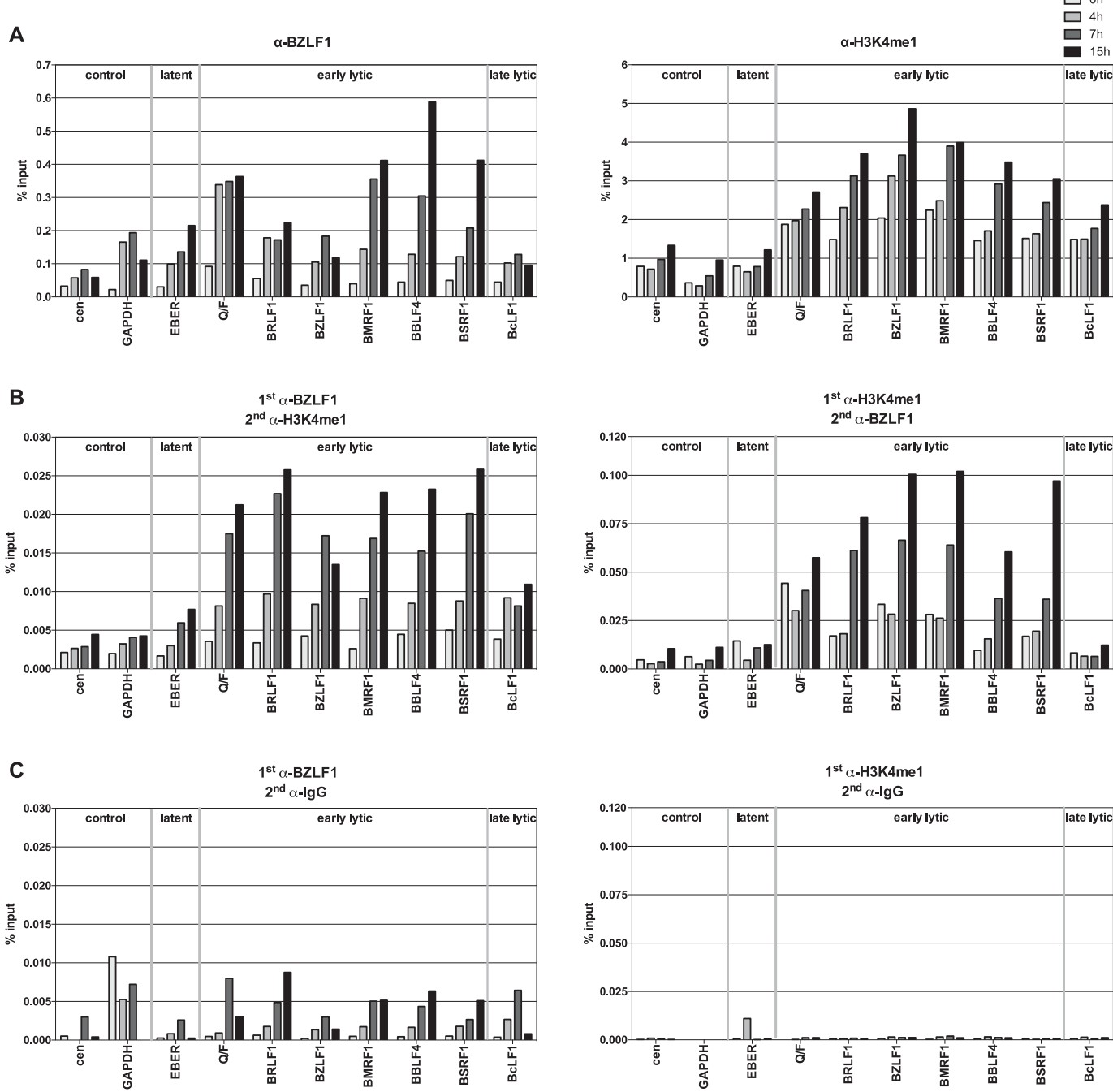

**Figure 2. BZLF1 and histone octamers co-occupy promoter sites of early lytic genes in vivo.**
**(A)** qPCR data of ChIP experiments with Raji p4816 cells are shown. Antibodies directed against BZLF1 or H3K4me1, and primer pairs specific for the indicated human (cen, GAPDH) or viral loci were used. Primer information can be found in Table S2. Mean values of three independent experiments are provided. **(B)** As panel A, but ReChIP experiments with sequential use of two different antibodies against either BZLF1 or H3K4me1. Right and left panel differ in the order of the antibodies used (indicated on top of the panels). Mean values of qPCR analysis of three independent ChIP replicates are provided. **(C)** As panel B, but with nonspecific IgG antibody as secondary antibody (indicated on top of the panels).

(Fig 3A). A 156-bp fragment of the BRLF1-coding sequence, which lacks ZREs and is not bound by BZLF1, served as negative control. All four 156-bp fragments, which had been fully CpG-methylated using a commercially available de novo CpG methyl transferase, were reconstituted into mononucleosomes by salt gradient dialysis using *Drosophila* embryo histone octamers (Krietenstein et al, 2012) (Fig S3A and B). Strep-tagged BZLF1 protein was expressed in HEK293 cells and purified under native conditions by Strep-Tactin affinity chromatography (Fig S3C and D) and quantified using bovine serum albumin as protein standard (Fig S3E).

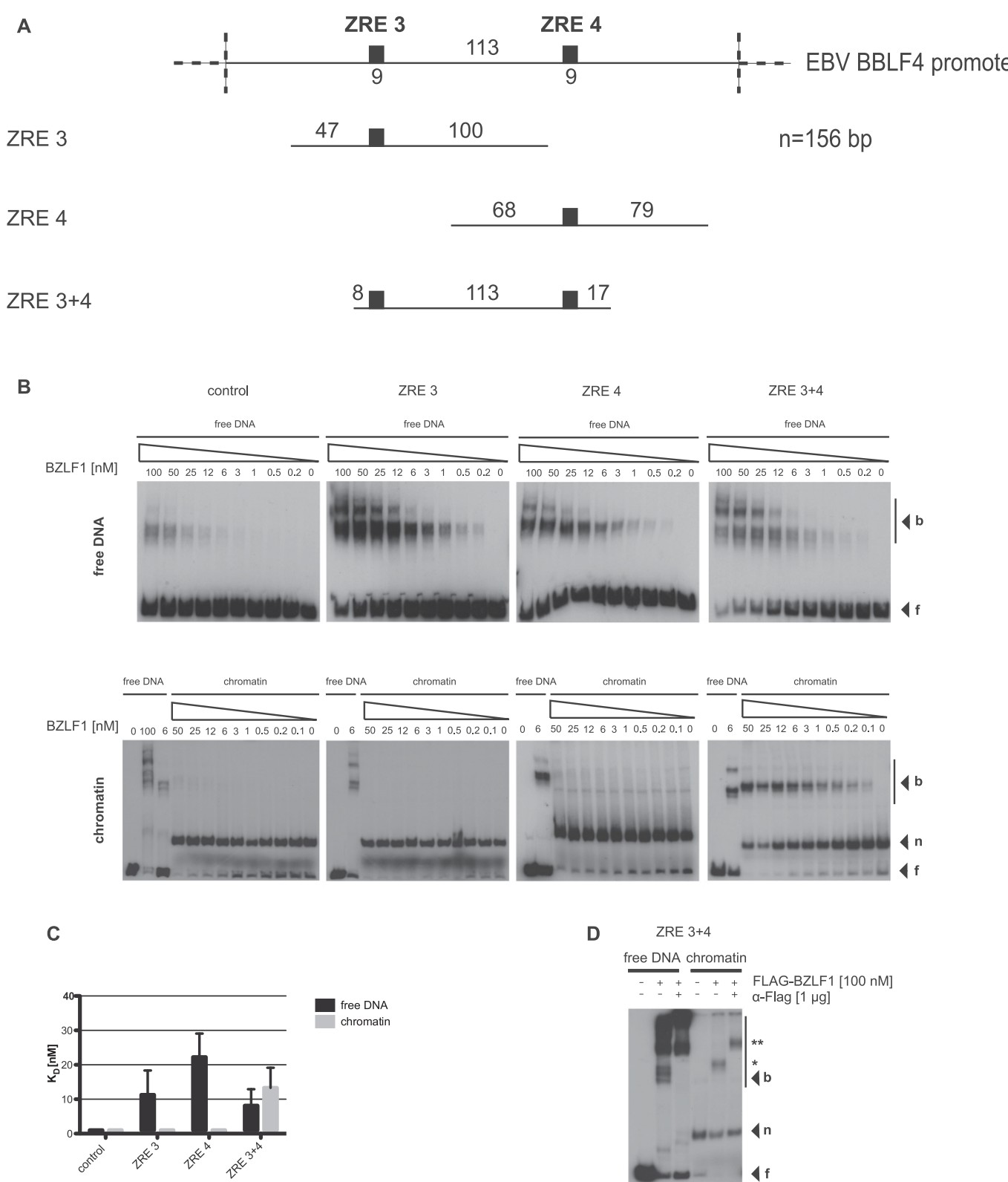

**Figure 3. BZLF1 binds mononucleosomes in vitro.**
**(A)** Top: schematics of the relative position of the two BZLF1-responsive elements ZRE 3 and ZRE 4 (black boxes) at the BBLF4 promoter. Numbers indicate DNA lengths in bp. Below: schematics of three different 156-bp fragments encompassing the indicated ZREs. Both ZREs contain a CpG motif that must be methylated for BZLF1 binding. **(B)** EMSAs for binding of BZLF1 (at indicated concentration) to DNA fragments as in panel A or to an EBV control region without ZRE that is not bound by BZLF1. DNA was either free or assembled into mononucleosomes by salt gradient dialysis (chromatin). The migration positions of free DNA (f), mononucleosomes (n), or the

EMSAs with these purified reagents (Fig 3B) allowed measuring BZLF1's binding to the four DNA fragments in their free (upper row of panels) or mononucleosomal (lower row of panels) states and the determination of the respective equilibrium dissociation constants (KD) (Fig 3C). BZLF1 bound with similar affinity (KD of ~10–20 nM) to the three ZRE-containing and histone-free DNA fragments consistent with previous experiments (Bergbauer et al, 2010) and independent of the number of ZREs. BZLF1 bound only weakly to the control fragment lacking a ZRE (Fig 3B), which is consistent with the widely observed, low-level but nonspecific DNA binding of transcription factors (Fried & Crothers, 1981). BZLF1's binding to free DNA resulted in several shifted bands, which is a common observation (Bergbauer et al, 2010).

In contrast, BZLF1's binding to mononucleosomal DNA differed for the three ZRE-containing fragments (Fig 3B, lower row of panels). BZLF1 did not bind to mononucleosomes without (control fragment) or with only one ZRE (ZRE 3 or ZRE 4). The well-studied yeast transactivator Pho4, which served as a negative control because it does not bind to nucleosomal sites (Venter et al, 1994), did not yield shifted bands with its binding site buried in mononucleosomes but did with free DNA (Fig S4). However, BZLF1 did bind to the ZRE 3+4 fragment, again with a KD of about 13 nM (Fig 3C). A truncated BZLF1 protein (aa 149–245) that lacks the activation but retains the DNA-binding domain, bound both free and mononucleosomal ZRE 3+4 fragments (Fig S5), indicating that the DNA-binding domain is sufficient to mediate the pioneer factor–like binding of BZLF1.

The binding of BZLF1 to the mononucleosomal ZRE 3+4 fragment (but not to fragments with single ZRE 3 or ZRE 4 motifs) suggested that BZLF1 requires at least two binding sites for stable binding to a nucleosome. As an alternative interpretation, two ZREs might be required to outcompete the histone octamer for binding such that the shifted complex migrated like a complex of BZLF1 with free DNA. We ruled out this latter possibility by comparing BZLF1 complexes with free and mononucleosomal ZRE 3+4 fragments run in parallel in the same gels (Fig 3B, bottom row, rightmost panel, and Fig 3D). The migration position of BZLF1 in complex with free DNA differed from that in complex with mononucleosomal DNA. An anti-FLAG antibody appropriate for binding FLAG-tagged BZLF1 supershifted the signals and unambiguously identified BZLF1 in both the free and the mononucleosomal DNA shift complexes (Fig 3D).

Yet another interpretation could be that BZLF1 did not necessarily require two ZREs for binding on a nucleosome, but that a ZRE had to be close to the entry and or exit points of the nucleosomal DNA rather than close to the dyad. It is known that nucleosomal DNA can undergo thermal "breathing" motions that transiently expose DNA sites close to the exit/entry points but much less frequently sites close to the dyad (Anderson & Widom, 2000). To investigate this possibility, we modified the ZRE 3 or ZRE 4 sequences in the ZRE 3+4 fragment by PCR mutagenesis such that we obtained two fragments with only one ZRE located at different positions relative to the entry/exit points termed ZRE 0+4 and ZRE 3+0 (Fig 4A). EMSAs with BZLF1 and these two constructs in

mononucleosomal forms demonstrated that BZLF1's binding to ZRE 0+4 was barely detectable (Fig 4B, middle panel), relatively strong to ZRE 3+0 (Fig 4B, right panel) and strongest to ZRE 3+4 (Fig 4B, left panel). From Hill plots (Fig 4C), we found again a dissociation constant of 13 nM for BZLF1 binding to the ZRE 3+4 mononucleosome compared with a KD of about 100 nM for ZRE 3+0. The dissociation constant could not be determined for ZRE 0+4 mononucleosomes. In contrast and as expected, the KD values of BZLF1 binding to the different free DNA fragments were in the range of 10–20 nM in three independent experiments (Fig S6).

We made use of the clearly detectable binding of BZLF1 to the single ZRE in the ZRE 3+0 fragment to ask if thermal "breathing" or nucleosomal phasing played a major role to this binding. To do so, we altered the position of the single ZRE in the ZRE 3+0 fragment relative to the original position of this ZRE by −5 nt, +10 nt, +15 nt, and +30 nt as shown in Fig 4A. With these four constructs, we repeated the EMSA analysis and observed robust BZLF1 binding to three of four mononucleosomes tested (−5 nt, +15 nt, and +30 nt), in all cases stronger than to the ZRE 0+4 mononucleosome (Fig 4D versus B). BZLF1 binding to ZRE 3+0 +10 nt (Fig 4A) was not detectable (Fig 4D, middle panel). This finding discredits a prominent role of DNA "breathing" for BZLF1's binding, especially given that the ZRE is more internal in the ZRE 3+0 +30 nt than in the ZRE 0+4 fragment. The fact that BZLF1 did not bind to ZRE 3+0 +10 nt (Fig 4D, middle panel) is reminiscent of ZRE 0+4 (Fig 4B, middle panel), because both binding sites are positioned similarly, that is, 18 and 17 nt from the distal ends (Fig 4A). Conversely, the DNA fragment likely lacks a strong nucleosome positioning sequence and is longer than 146 bp such that in the absence of a linker histone, the DNA molecule might slide somewhat around the histone core. Nevertheless, the data suggested that nucleosomal phasing might be a critical determinant and that the single ZRE when positioned on the surface of the histone octamer likely enables BZLF1 binding (Fig 4D).

Taken together, we conclude that the properties and the position of a given ZRE (ZRE 3 versus ZRE 4) and, to a much larger degree, the cooperation between two ZREs (ZRE 3+4 construct) support BZLF1's binding to a nucleosomal site.

## BZLF1 interacts with cellular chromatin-remodeling enzymes

In our in vitro experiments with reconstituted nucleosomes, we did not observe a histone loss or a disassembly of the nucleosome upon BZLF1's binding because shifted bands characteristic of a BZLF1-free DNA complex (Fig 3D) or an increase of free DNA (Fig 3B, bottom row, rightmost panel) were not detected. It thus appeared that BZLF1's binding to chromatin and the ejection of nucleosomes in vivo are two distinct but possibly linked processes.

We hypothesized that in vivo BZLF1 might first bind at ZREs in promoter elements of lytic target genes on top of the nucleosomes without ejecting them, but then recruit cellular chromatin-remodeling enzymes that would mediate the loss of histones.

complexes with BZLF1 (b) are indicated on the right of the autoradiographs. **(C)** Quantification of equilibrium dissociation constants (KD) from EMSA experiments as in panel B. If error bars are provided, the average and SD of three independent experiments are shown. **(D)** EMSA ("super shift assay") with FLAG-tagged BZLF1 (FLAG-BZLF1), free or mononucleosomal ZRE 3+4 DNA and anti-FLAG (α-FLAG) antibody as indicated. The migration positions of the FLAG-BZLF1/mononucleosome and the FLAG-BZLF1/α-FLAG/mononucleosome complexes are indicated on the right by one (*) and two asterisks (**), respectively.

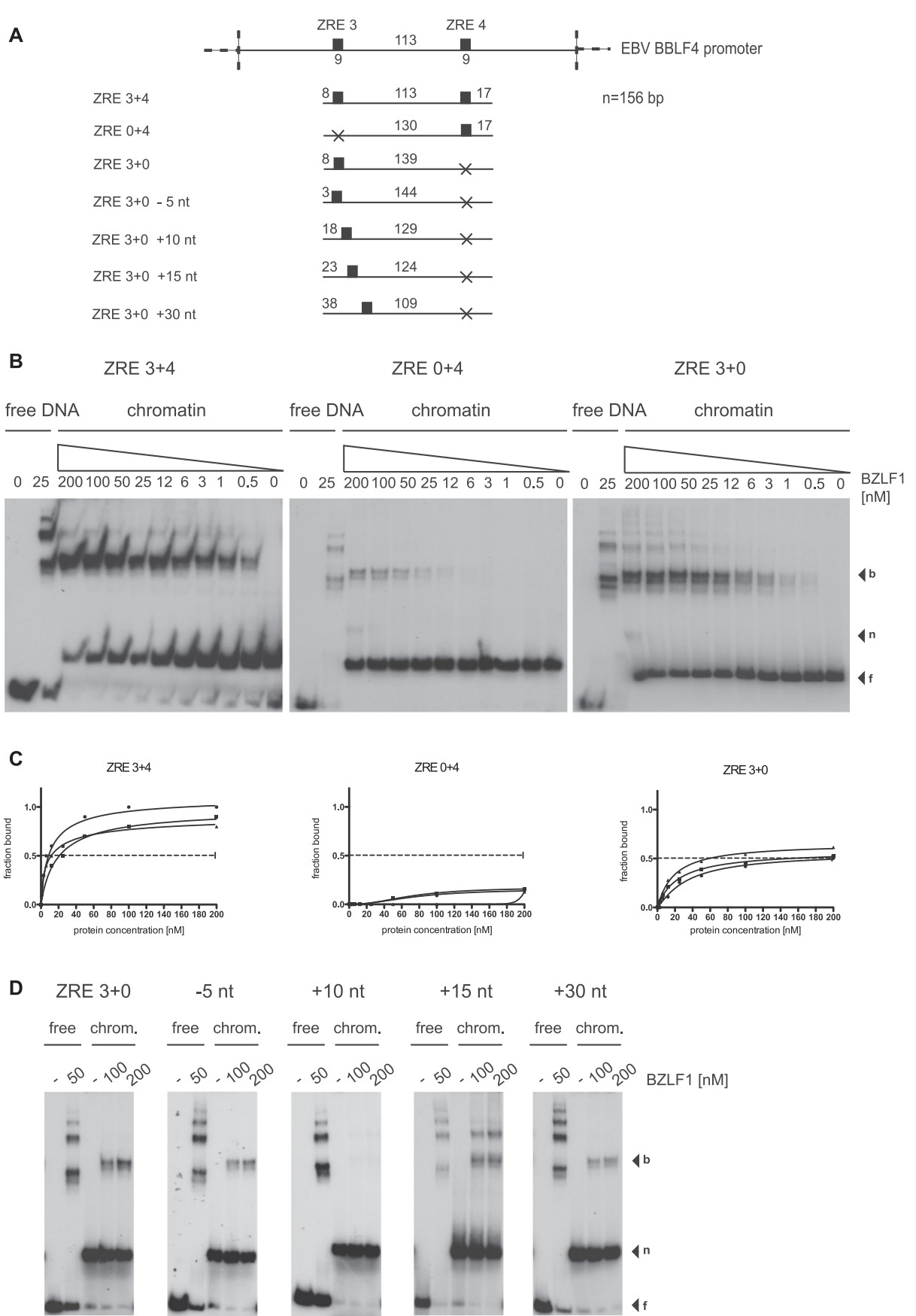

With a candidate approach, we analyzed whether BZLF1 interacted with members of the chromatin remodeler families, ISWI and INO80. We co-immunoprecipitated BZLF1 and the catalytic ATPase subunits SNF2h (encoded by the gene *SMARCA5* and corresponding to the paradigmatic *Drosophila* ISWI ATPase [Flaus et al, 2006]) or INO80 (Shen et al, 2000). The chromatin remodelers were both expressed at endogenous levels in Raji cells, whereas BZLF1 was expressed in cells stably transfected with our doxycycline-inducible expression system (Fig S1) encoding Strep-tagged BZLF1 full-length protein (aa 1–245) or only the BZLF1 bZIP domain (aa 175–236). After overnight induction, the cell lysates were treated with benzonase and DNase I before immunoprecipitation to eliminate nucleic acid–mediated recovery of the factors. Tagged BZLF1 was immunoprecipitated on streptavidin beads, and co-precipitation of chromatin remodelers was determined by subsequent Western blotting (Fig 5A). This approach revealed interactions of full-length BZLF1 with endogenous SNF2h and INO80 protein but not with CHD4. The BZLF1 bZIP domain alone, that is, BZLF1 lacking its TAD and the ultimate carboxy terminus, interacted with SNF2h but not with INO80 (Fig 5A). These results suggested that BZLF1 interacts with subunits of at least two cellular chromatin remodeler families possibly to recruit them to lytic gene promoters and support EBV's lytic reactivation.

### Different BZLF1 domains mediate the interaction with SNF2h versus INO80

The differential binding of the bZIP constructs to SNF2h versus INO80 indicated that different domains of BZLF1 might mediate these interactions. We extended our co-immunoprecipitations (Co-IPs) to include more BZLF1 derivatives (Fig 5B). We transiently transfected HEK293 cells with the BZLF1 constructs, which were co-expressed with the GFP-tagged chromatin remodeler ATPase subunits, SNF2h or INO80. The BZLF1 expression plasmids were adjusted to obtain similar protein levels (Fig 5C). Cell lysates were again treated with benzonase and DNase I, and immunoprecipitations were carried out with GFP binder–coupled Sepharose beads, and the co-precipitated BZLF1 was detected by Western blotting using the antibody directed against a motif within BZLF1's bZIP domain. The immunoblots (Fig 5D; controls in Fig S7) demonstrated that SNF2h interacted with all BZLF1 derivatives tested (upper panel) probably identifying aa175-236 of BZLF1 as the domain responsible for SNF2h interaction. In contrast, the intact TAD of BZLF1 was essential to bind INO80 or components of the INO80 remodeler complex (Figs 5D and S7), a feature, which is in agreement with the presumed functions of activation domains of DNA-binding transcription factors in general. Therefore, we put our focus on INO80 in most of the following experiments, in which we investigated the functional role of chromatin remodelers in lytic viral reactivation.

### BZLF1 supports the recruitment of INO80 to viral DNA

The transactivating domain of BZLF1 appeared to interact with INO80 or components of the INO80 remodeler complex. We wondered whether BZLF1 might as well recruit this chromatin remodeler to viral chromatin, where it could induce local histone loss as seen in Fig 1B, for example. It is technically very challenging to perform convincing ChIPs with remodeler complexes in yeast, drosophila, or mammalian cells probably because these multicomponent complexes are large and presumably make only transient contacts with DNA-binding factors and chromatin. According to a recent article (Zhou et al, 2016), we performed ChIP experiments with an INO80-specific antibody and chromatin obtained from non-induced Raji p4816 cells and cells induced with doxycycline for 6 and 15 h (Fig 6). PCR primer pairs were selected that covered five early lytic viral promoters known to be bound by BZLF1 and three control loci in cellular chromatin, where BZLF1 does not bind (data not shown). In four independent experiments and only at viral promoters, we found a modest but reproducible increase of INO80 when the expression of BZLF1 was induced for 15 h (Fig 6). This finding supports our notion suggesting that BZLF1 possibly recruits the INO80 remodeler complex to these lytic viral promoters.

### High BZLF1 levels induce open chromatin at BZLF1-binding sites in Raji EBV DNA

We postulated that the site-specific binding of BZLF1 would induce the subsequent opening of these loci indicative of BZLF1's recruitment of cellular chromatin remodelers. To address this point, we used the Omni-ATAC-seq technology (Buenrostro et al, 2013; Corces et al, 2017) that can provide information about accessible regions of chromatin with base-pair resolution. A hyperactive transposase preferentially inserts adapter sequences into open chromatin, which act as primers to generate next generation sequencing libraries. We performed ATAC-seq experiments with non-induced and doxycycline-induced Raji p4816 and Raji p5694 cells (Fig S1) and analyzed the EBV genome-wide chromatin accessibility under these conditions. In non-induced Raji cells, only single discrete sites of open chromatin were found with the exception of a wider region at around *oriP* and the EBER locus at the left end of genomic EBV DNA (Fig 7A, tracks 1 and 4). In non-induced cells, many but not all accessible sites of open chromatin co-located with CTCF-binding sites, which might be relevant for the structure of EBV genomic DNA (Fig 7A and left panel of Fig 7B, tracks 1 and 6). Upon doxycycline-mediated expression of AD-truncated BZLF1 in Raji p5694 cells, the situation did not change, but the expression of full-length BZLF1 in Raji p4816 cells for 15 h caused a dramatic increase of open chromatin in EBV DNA (Fig 7, track 3). We also analyzed the EBV genome-wide binding of full-length BZLF1 with ChIP-seq and the BZLF1-specific BZ1 antibody in Raji p4816 cells 15 h after adding doxycycline (Fig 7A, track 2). We found that the ATAC-seq

**Figure 4. BZLF1 shows cooperative binding to the nucleosomal core in vitro.**
**(A)** Shown are the DNA templates used for the functional analysis of the two BZLF1-responsive elements ZRE 3 and ZRE 4 in the promoter of BBLF4 as in Fig 3A. **(B)** EMSA results of BZLF1 and the DNA templates ZRE 3+4, ZRE 0+4, and ZRE 3+0 suggested a cooperative binding of BZLF1 to ZRE 3 and ZRE 4. The ZRE 3+4 template was robustly bound by BZLF1, which interacted less efficiently with ZRE 3+0 and barely with ZRE 0+4. **(C)** Individual Hill slope curves of BZLF1 binding to the mononucleosomal DNA templates ZRE 3+4, ZRE 0+4, and ZRE 3+0 show the result of three independent experiments. **(D)** The position of the ZRE 3 motif within the DNA template ZRE 3+0 was altered as illustrated in (A). BZLF1 was competent to bind its ZRE 3 site in two more proximal positions (ZRE +15 nt and +30 nt) within the nucleosomal core.

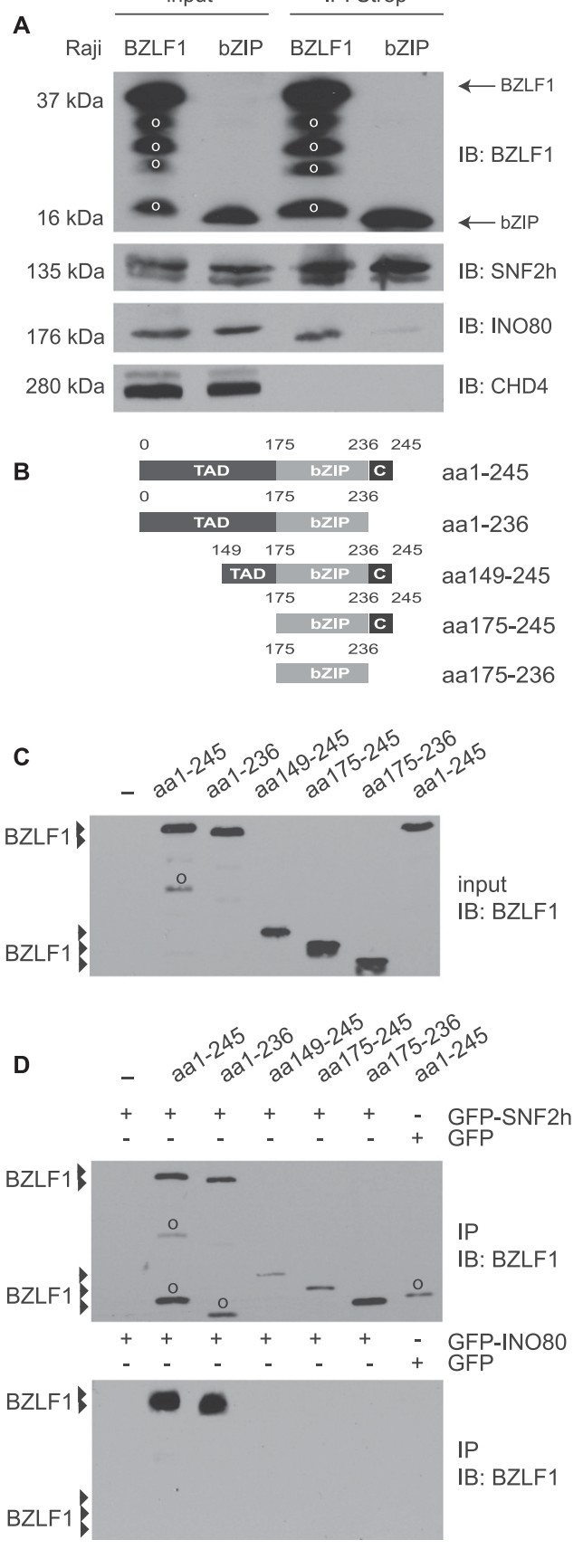

and ChIP-seq patterns were mostly congruent comparing track 2 with track 3. Fig 7B shows three examples, one of which (the left BZLF1 peak in the left panel) indicates an exception of this common observation.

Next, we asked whether an alignment of the many BZLF1 ChIP-seq peaks in EBV DNA with the coverage of our ATAC-seq data might be informative. Towards this end, we employed the peak caller MACS2 (Feng et al, 2012) and identified 67 BZLF1-binding sites in doxycycline-induced Raji p4816 cell DNA (Fig S8A). We calculated the average BZLF1 peak coverage using HOMER (Heinz et al, 2010) as schematically shown Fig S8B but noticed that the density of BZLF1-binding sites in genomic EBV chromatin is often high. As a consequence, the flanks of neighboring BZLF1 peaks overlap such that the average BZLF1 peak in EBV DNA is broad very much in contrast to host chromatin in Raji cells, where more than $10^5$ individual BZLF1 sites are mostly isolated and widespread (Buschle et al, 2019 Preprint).

Next, we calculated the average ATAC-seq peak coverage as schematically shown Fig S8C. The metaplot in panel A of Fig 8 displays the ATAC-seq coverage of open chromatin projected onto the center of the 67 BZLF1-binding sites in Raji EBV DNA within a window of ±400 nt. Three conditions, non-induced Raji p4816 and both non-induced and induced Raji p5964 cells were almost indistinguishable and showed a flat and very low ATAC-seq coverage on average. Addition of doxycycline for 15 h caused a dramatic increase of chromatin accessibility in Raji p4816 cells with a hilltop pattern peaking almost at the center of the average BZLF1 peak (Fig 8A). The corresponding box plot (Fig 8B) confirms this observation. Four heat maps (Figs 8C, D, and S9) show the individual ATAC-seq coverage at all 67 BZLF1-binding sites in a ranked hierarchical order reflecting the four cellular conditions. Only the induced expression of full-length BZLF1 caused a change in the global heatmap patterns, indicating an increase in accessible EBV chromatin that is concurrent with the center of the 67 identified BZLF1-binding sites (Fig 8D).

**Figure 5.  BZLF1 interacts with the core subunits of the cellular chromatin remodelers SNF2h and INO80 in vivo.**
**(A)** Raji cell lines were stably transfected with tetracycline-regulated expression plasmids encoding Strep-tagged BZLF1 full-length or bZIP protein (Fig S1) consisting of aa 175 to aa 236 with the DNA-binding and dimerization domains of BZLF1. After treatment with benzonase and DNase I, the Strep-tag fusion proteins of lytically induced cells were captured with Streptavidin beads. Co-precipitated proteins were analyzed with antibodies directed against SNF2h, INO80, and CHD4. In the immunoblot detecting BZLF1 and bZIP (top panel), 0.5% of the total protein lysate was loaded as "input" per lane; in each of the two lanes labeled "IP:Strep," 10% of the immunoprecipitated material was loaded. In the immunoblots detecting SNF2h, INO80, or CHD4, 1% of the total protein lysate was loaded as "input" per lane; in the lanes labeled "IP:Strep," 90% of the immunoprecipitated material was loaded per lane. "o" indicates signals from proteolytic degradation. **(B)** Shown are the modular structures of truncated BZLF1 variants. TAD indicates the TAD of BZLF1. **(C)** Protein expression of the truncated BZLF1 variants (see panel B) in HEK293 cells was analyzed by immunodetection with the BZLF1-specific BZ1 antibody. **(D)** HEK293 cells were co-transfected with expression plasmids encoding BZLF1 variants and GFP-tagged chromatin remodeler ATPase subunits SNF2h or INO80. The cell lysates were treated with the enzymes benzonase and DNase I before immunoprecipitations of the GFP-tagged chromatin remodelers with GFP-binder beads. The analysis of input and immunoprecipitated material was performed by Western blot detection with the BZ1 antibody directed against BZLF1. "o" indicates signals from proteolytic degradation.

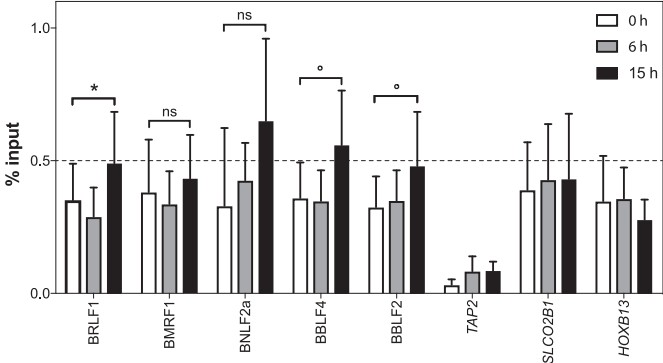

**Figure 6. INO80 is enriched at BZLF1-regulated early lytic EBV promoters.**
Chromatin from Raji p4816 cells induced with doxycycline to initiate the expression of BZLF1 for 0, 6, or 15 h were used to perform ChIP experiments with an INO80 antibody. The recovered DNAs were quantified by qPCR with suitable primer pairs (Table S2). We compared the input versus ChIPed DNAs expressed as "% input." Shown is the analysis of five different early lytic promoter regions. Three cellular loci, where BZLF1 does not bind, served as negative controls. Mean and SD values from four different experiments are provided. The results with non-induced (0 h) and induced (15 h) samples were analyzed with the paired t test, and significance levels were defined as *P < 0.05 and °P < 0.1; ns, not significant.

### shRNA-mediated knock-down of INO80 reduces transcriptional reactivation of certain early lytic genes of EBV

Next, we asked whether INO80 levels might be important to activate viral lytic genes upon expression of BZLF1. We engineered lentiviruses to stably express shRNAs directed against INO80 transcripts in Raji p4816 cells (Fig S10A and B) and tested the timely expression of selected viral lytic genes upon addition of doxycycline by qRT-PCR.

As can be seen in Fig S11A, two shRNAs efficiently reduced the steady-state levels of INO80 protein in Raji cells. We next analyzed the transcriptional activation of four early viral genes (BMRF1, BNLF2a, BRLF1, and BBLF4) in three different Raji p4816 cell lines stably transduced with shRNA_INO80_1, shRNA_INO80 _2, or a non-targeting shRNA_nt control using lentiviral vectors (Fig S11A and B). The knock-down of INO80 resulted in a reduced activation of BMRF1 and BNLF2a 8 h post induction (Fig S11B). In contrast to these two "early responding" genes, BRLF1 and BBLF4, which have considerably slower kinetics of induction, showed a very modest reduction of their transcript levels 15 h post induction only (Fig S11B). We cannot exclude possible indirect effects because INO80 regulates a vast array of genes, but this experiment suggests that INO80 might play an important functional role in the early phase of viral reactivation at certain viral promoters of early lytic genes.

### siRNA-mediated knock-down of INO80 inhibits de novo synthesis of virus

As BZLF1 is the crucial trigger for viral reactivation and interacts with at least two cellular chromatin remodelers, we hypothesized that at least one of them should be necessary for lytic induction. To test this hypothesis, we used an siRNA knock-down strategy to assess the roles of SNF2h or INO80 in 2089 EBV HEK293 cells (Delecluse et al, 1998). Upon transient transfection of a BZLF1 expression plasmid (Hammerschmidt & Sugden, 1988), this 2089 EBV HEK293 producer

cell line releases infectious virus, which can be quantified by assaying infected, GFP-positive Raji cells by flow cytometry (Steinbrück et al, 2015).

The 2089 EBV HEK293 cells were treated for three days with siRNA pools targeting the *SMARCA5* gene encoding SNF2h or the *INO80* gene or, for control, with a non-targeting siRNA. The respective knock-down efficiencies were assessed by Western blotting (Fig 9A). In these siRNA-treated cells virus synthesis was initiated by transient co-transfection of expression plasmids encoding BZLF1 together with gp110/BALF4 as described (Neuhierl et al, 2002). Expression of gp110/BALF4 increases virus infectivity by about a factor of 10 (Neuhierl et al, 2002). Three days after plasmid co-transfection, cell supernatants were collected and defined volumes were used to infect Raji cells. The fractions of GFP-expressing Raji cells were determined by flow cytometry after three additional days such that the virus concentrations could be calculated (Fig 9B).

Steady-state protein levels of SNF2h or INO80 were modestly reduced after three days of siRNA treatment (Fig 9A). Cells treated with non-targeting siRNAs or with an SNF2h-specific siRNA pool (Fig S10C) did not differ significantly in the levels of released, infectious EBV (Fig 9B). However, cells treated with an INO80-specific siRNA pool released significantly fewer viral particles (Fig 9B). This observation is notable given the only modest diminution of INO80 by the siRNA treatment (Fig 9A). None of the siRNA pools directed against SNF2h or INO80 had an adverse effect on cell viability (Fig S10D), suggesting that the reduced virus synthesis after siRNA knock-down is a specific and, probably, INO80-related effect.

Together, these results support an important role for INO80 and for BZLF1's acting as a pioneer factor in EBV lytic activation.

## Discussion

EBV takes advantage of the host cell's epigenetic machinery to establish a stable latent infection. Upon infection, the viral DNA is epigenetically naïve, that is, free of histones and devoid of methylated CpG dinucleotides (Kintner & Sugden, 1981; Fernandez et al, 2009; Kalla et al, 2010). In the course of establishing the latent phase, the host cell's epigenetic machinery compacts the viral DNA into nucleosomal arrays, introduces repressive histone modifications, and initiates the methylation of most viral CpG dinucleotides (Kalla et al, 2010). As a consequence, viral promoters, with the exception of those of a few latent genes, are silenced during EBV's latent phase. Densely positioned nucleosomes, repressive histone marks introduced by Polycomb proteins, and extensive DNA methylation keep the virus in a strictly latent, dormant mode (Ramasubramanyan et al, 2012; Woellmer et al, 2012).

EBV can escape from latency and enter the lytic, productive phase when its host B cells terminally differentiate to plasma cells (Laichalk and Thorley-Lawson, 2005). In the lytic phase, the loss of nucleosomes increases the accessibility of viral DNA to binding transcription factors. The removal of repressive and the gain of active histone marks reactivate the promoter regions of early viral lytic genes, enabling the virus to replicate its DNA, express late viral genes, and produce viral progeny to infect new B cells (Hammerschmidt, 2015 for a recent review).

The viral transcription factor BZLF1, which is induced upon terminal plasma cell differentiation, is the switch triggering a

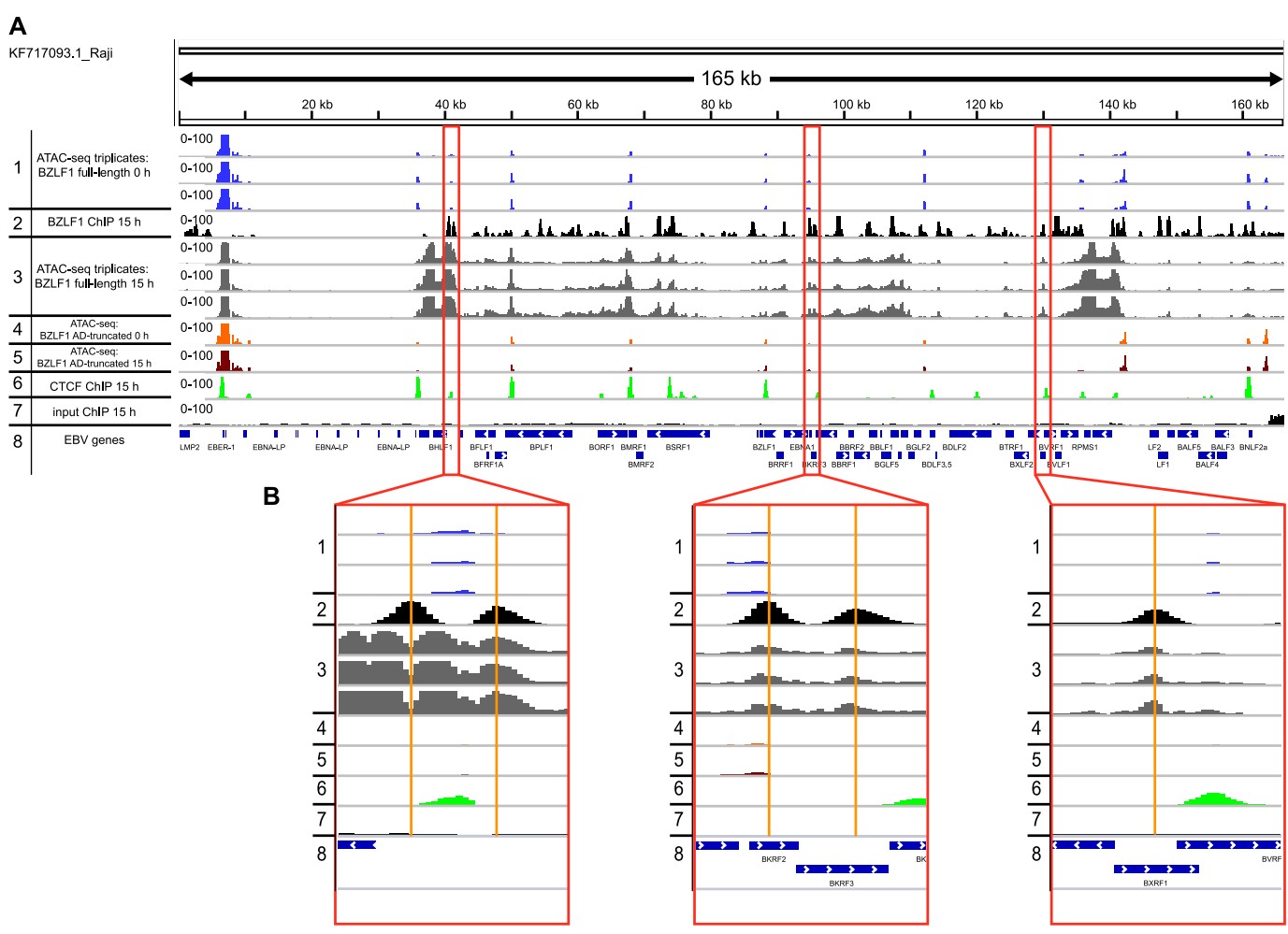

**Figure 7. EBV genome-wide ATAC-seq coverage in non-induced and induced Raji p4816 cells.**
**(A)** The normalized coverage of three independent ATAC-seq experiments are shown in non-induced Raji p4816 cells (track 1) and cells induced for 15 h (track 3). The reads from six ATAC-seq experiments are aligned on the complete Raji EBV genome (KF717093.1) together with normalized ChIP-seq data obtained with BZLF1- or CTCF-specific antibodies (tracks 2 and 6, respectively). Additional controls include ATAC-seq reads from AD-truncated Raji cells before and after induction for 15 h (tracks 4 and 5, respectively), which do not differ and are indistinguishable from ATAC-seq reads found in non-induced Raji p4816 cells (track 1). The input control (track 7) pairs with ChIP-seq experiments at 15 h post induction shown in tracks 2 and 6 and indicate the low and even level of mappable reads before ChIP. Track 8 provides the positions of selected EBV genes in the 165-kb Raji genome. **(B)** The three panels provide individual examples of BZLF1 peaks in Raji cells that appear after doxycycline induction of full-length BZLF1 (track 2) and the concomitant increase in chromatin accessibility (track 3). The tracks are annotated according to tracks in panel A.

transition from the latent to the lytic phase. First, BZLF1 induces the expression of several early lytic EBV genes by binding sequence-specifically to ZREs in their promoter regions. Many ZREs need to contain 5-methyl cytosine residues to permit BZLF1's binding, and thus, CpG methylation of viral DNA is a prerequisite to express certain essential, early lytic genes (Kalla et al, 2010, 2012). Second, BZLF1 enables viral DNA replication. It binds to the lytic origin of DNA replication and promotes the recruitment of components of the viral DNA replication machinery to initiate lytic viral DNA replication (Schepers et al, 1993). Third, BZLF1 directly or indirectly causes the loss of nucleosomes in the promoter regions of viral lytic genes, which correlates with their expression (Hammerschmidt, 2015).

We and others have hypothesized that the efficient reactivation of silenced, inactive viral chromatin is forced by a presumed pioneer function of the BZLF1 transcription factor (Woellmer et al,

2012), chromatin alterations (Adamson & Kenney, 1999; Zerby et al, 1999), and/or the additional recruitment of chromatin remodelers (Woellmer et al, 2012). Our results, both in vivo (Fig 2) and in vitro (Figs 3 and 4), show that BZLF1 can bind mononucleosomal DNA in promoter regions of early lytic genes known to be regulated by BZLF1. We have also shown that BZLF1 can bind ZREs in nucleosomes close to the nucleosome dyad (Fig 4D). The nucleosome-binding activity is encoded within the C-terminal part of BZLF1 encompassing the bZIP domain and does not depend on BZLF1's TAD (Fig S5). BZLF1's binding does not eject nucleosomes in vitro (Fig 3B and D) in the absence of other molecular machines. This observation is in line with our initial data (Fig 1B) indicating that the binding of BZLF1 precedes a decrease in nucleosomal occupancy at early lytic promoters by hours.

Transcriptional activation of lytic viral genes (e.g., BMRF1 and BRLF1 in Fig S11B) appears to precede nucleosome ejection as

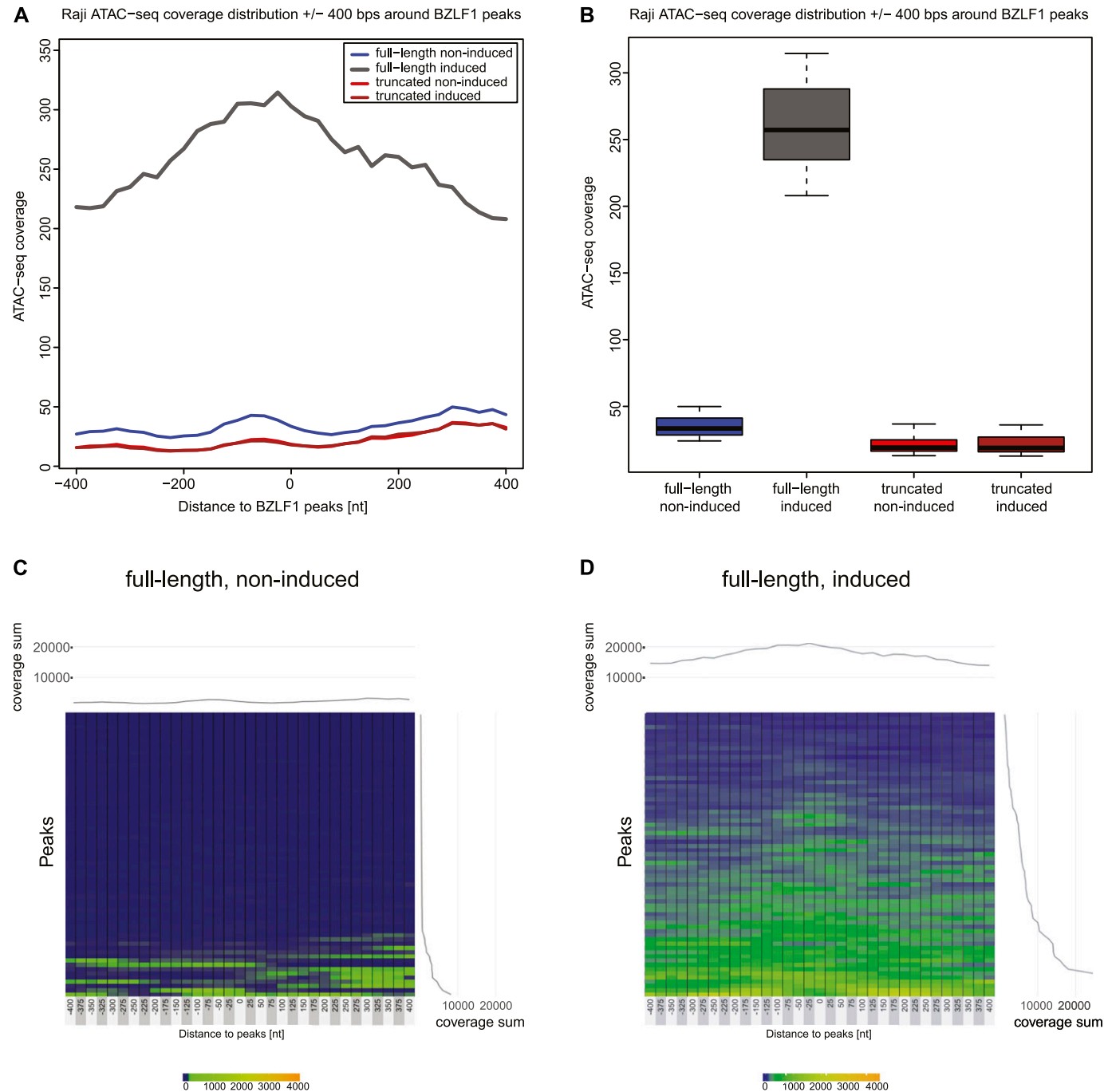

**Figure 8. ATAC-seq coverage at the 67 BZLF1-binding sites in Raji EBV chromatin.**
**(A)** The panel shows the metaplot of the average ATAC-seq coverage at BZLF1 peaks in Raji EBV DNA. The plot covers a ±400-nt range centered at the maximal heights of 67 BZLF1 peaks identified by the MACS2 peak caller in ChIP-seq experiments with an antibody directed against BZLF1. Four ATAC-seq conditions are indicated comparing non-induced Raji p4816 cells (thin blue line; full-length, non-induced) and cells induced for 15 h (thick grey line; full-length, induced) as well as AD-truncated Raji cells before and after induction (thin red and brown lines; truncated, non-induced and induced, respectively). **(B)** The boxplot quantifies the average data shown in panel A. **(C, D)** The two heatmaps summarize the individual ATAC-seq coverage at the identified 67 BZLF1 ChIP-seq peaks in Raji EBV chromatin arranged in hierarchical order. The left (C) and right (D) panels illustrate the situation in non-induced Raji p4816 cells (full-length) and cells induced for 15 h (full-length), respectively. The data show the mean of three independent ATAC-seq experiments. Heatmaps of two sets of ATAC-seq data from non-induced and induced AD-truncated Raji cells are provided in Fig S9.

shown in Fig 1B at 15 h post induction. This lag is probably due to the fact that the Raji cell line contains about 15–20 copies of viral genomes per cell. The loss of histones can be detected, only, when a considerable percentage of nucleosomes is removed given the

background of the many EBV genome copies per cell. It is unlikely that they all become activated and respond synchronously to the expression of BZLF1. The knock-down of INO80 reduces transcriptional activation of the BMRF1 and BNLF2a genes 8 h post induction,

whereas moderate repressive effects can be seen at the BRLF1 locus 15 h post induction (Fig S11B). This observation does not prove but seems to support our working hypothesis. Together with our in vitro and in vivo data in Figs 3 and 4, respectively, in Figs 7 and 8, it appears that BZLF1 binds to nucleosomal DNA and concomitantly recruits chromatin remodelers such as INO80. Chromatin remodelers such as INO80 will mobilize nucleosomes and evict them eventually to revert epigenetic silencing and promote gene activation.

Co-IPs suggest the in vivo interactions of BZLF1 with the ATPase subunits SNF2h and INO80 of the chromatin remodeler families ISWI and INO80, respectively (Fig 5). In HSV-1 VP16, BZLF1's functional counterpart, regulates the lytic reactivation process. VP16's TAD recruits general transcription factors and chromatin cofactors, including chromatin-remodeling enzymes to sites in viral promoters, dramatically reducing their histone occupancy (Neely et al, 1999; Herrera & Triezenberg, 2004). The ATPase subunit SNF2h has been reported previously to promote HSV-1 immediate-early gene expression as well as replication and might also interact with VP16 (Bryant et al, 2011).

Similarly, the INO80 remodelers are able to slide nucleosomes, exchange histones, regulate transcription, and are involved in DNA repair and cell cycle checkpoint control (Shen et al, 2000; Tsukuda et al, 2005; van Attikum et al, 2007). Whereas INO80 has not been implicated in the reactivation of herpes viruses, the EBV nuclear proteins EBNA-LP and EBNA2, which are necessary for lymphoblastoid cell line growth and survival, have been found associated with an INO80 remodeler complex (Portal et al, 2013). Our results indicate that BZLF1's TAD interacts with the INO80 ATPase subunit (Fig 5D) and likely recruits the entire remodeler complex to viral chromatin (Fig 6), where we envision that it mobilizes histone octamers and disrupts the nucleosome-dense regions of the viral promoters upon lytic reactivation. INO80 knock-down experiments support this critical role (Fig S11B).

Recruitment of INO80 via BZLF1's TAD is consistent with our identification of a subgroup of ZREs in viral DNA with a higher than average nucleosome occupancy during latency. Nucleosome loss and the formation of hypersensitive sites at these ZRE elements was only observed with full-length BZLF1 protein but not with the bZIP domain lacking BZLF1's TAD (Fig 2C, bottom panel, in Woellmer et al (2012)). Our ATAC-seq data in Figs 7 and 8 nicely recapitulate these previous findings and highlight the importance of the TAD of BZLF1.

Recent reports support the view that pioneer transcription factors can penetrate epigenetically silenced chromatin to recognize and bind their DNA-binding sites activating the associated genes (Zaret & Mango, 2016). The cell type–specific distribution of histone marks plays an important role in the recruitment of pioneer factors, as some are capable of reading them. For example, the recruitment of FoxA to enhancers depends on epigenetic changes of enhancer hallmarks (Sérandour et al, 2011). The nucleosome-binding activity of FoxA is facilitated by the histone marks H3K4me1 and H3K4me2 but not by histone acetylation (Cirillo & Zaret, 1999; Lupien et al, 2008). FoxA can even favor H3K4me2 deposition (Smale, 2010). The pioneer factor PBX1 is also capable of reading specific epigenetic signatures such as H3K4me2 (Berkes et al, 2004; Magnani et al, 2011a), and PU.1 reprograms the chromatin landscape through the induced deposition of H3K4me1 (Heinz et al, 2010). The viral transcription factor BZLF1 preferentially interacts with DNA motifs that contain methylated CpG dinucleotides, but might use other epigenetic modifications as well.

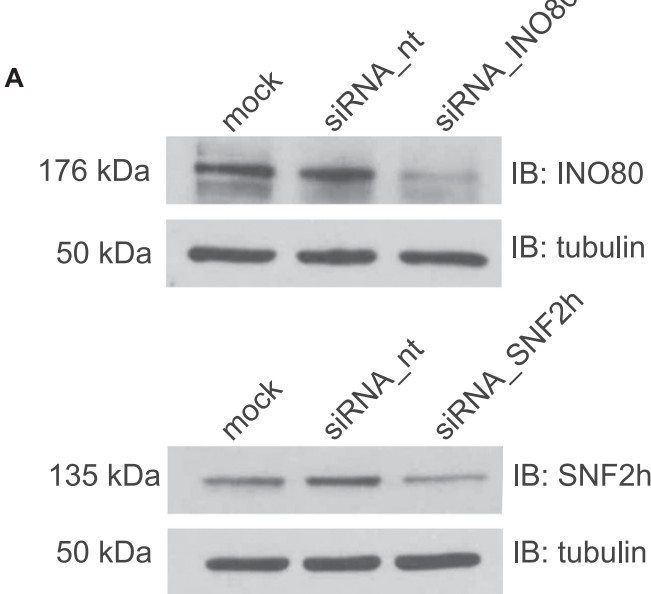

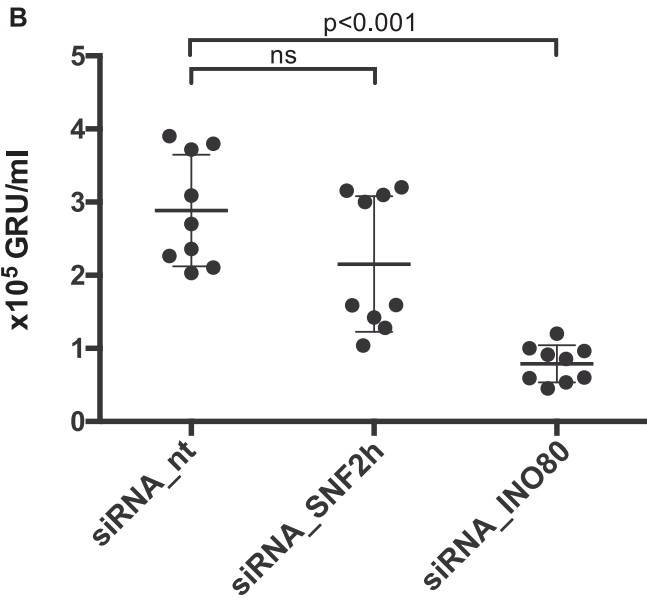

**Figure 9. siRNA knock-down of SNF2h and INO80 subunits reduce virus de novo synthesis.**
**(A)** Knock-down efficiencies of the SNF2h and INO80 ATPase subunits in 2089 EBV HEK293 cells were analyzed by immunoblotting. Tubulin served as a loading control. One representative experiment out of three is shown. Additional information can be found in Fig S10. **(B)** Quantification of virus concentrations released after lytic induction of 2089 EBV HEK293 cells indicate a reduction of virus synthesis after INO80 knock-down. Mean and SDs from nine independent experiments are shown. P values of an unpaired t test corrected with the Sidak–Bonferroni method are shown. GRU: green Raji units. ns, not significant.

The viral transcription factor BZLF1 belongs to the basic leucine-zipper (bZIP) family of transcription factors and contains a variant of the leucine zipper motif responsible for the coiled-coil structure (Farrell et al, 1989; Chang et al, 1990; Lieberman and Berk, 1990). BZLF1 forms

homodimers and binds DNA motifs via its two long bZIP helices (Petosa et al, 2006). The closest relative of BZLF1 is the cellular c-Fos/c-Jun heterodimer AP1 transcription factor (Farrell et al, 1989). Its binding to nucleosomal DNA is reduced compared with free DNA sequence motifs, but nucleosomal DNA binding severely affects the structure of the underlying nucleosome, which can facilitate the subsequent binding of additional transcription factors (Ng et al, 1997). It, thus, appears that BZLF1 has optimized this fundamental function of AP-1 transcription factors to support EBV's escape from repressed chromatin.

EBNA1 was also proposed to have similarities with the paradigmatic pioneer factor FoxA1 in a recent review (Niller & Minarovits, 2012), but to our knowledge, our biochemical and functional data identify BZLF1 as a bona fide pioneer factor of EBV. Two domains in EBNA1 mimic the AT-hooks of certain cellular high-mobility group proteins (Hung et al, 2001; Altmann et al, 2006) and promote the mobility of the linker histone H1 indicative of an EBNA1-intrinsic remodeling function, which is independent of cellular chromatin remodelers (Coppotelli et al, 2013).

The pioneer factor BZLF1 could share certain functions with its cousin, VP16 of HSV. Herpes simplex virus DNA is not (Leinbach & Summers, 1980; Muggeridge & Fraser, 1986) or only selectively associated with nucleosomes during lytic infection (Oh et al, 2015 and references therein), and partially conflicting data suggest that either histone chaperones (Oh et al, 2012), chromatin modifying co-activators (Herrera & Triezenberg, 2004), or chromatin remodelers (Neely et al, 1999) are responsible for the lack of histones on Herpes simplex DNA. Interestingly, the TAD of VP16 was found to associate directly with members of the SWI/SNF remodeling complex (Neely et al, 1999) activating in vitro transcription from a nucleosomal template (ibid). This latter finding is reminiscent of BZLF1's TAD interacting with the chromatin remodeler INO80 (Fig 5D), which appears to play an important role for efficient lytic activation of EBV (Fig 9).

It has also emerged from recent work (Soufi et al, 2015) that key factors in cellular reprogramming to yield induced pluripotent stem cells share critical functions with pioneer factors. Our current findings suggest that EBV has acquired this principle and puts it to use with its BZLF1 factor to reprogram viral latent chromatin within hours and to promote escape from latency.

Taken together, our experiments suggest that BZLF1 is a bona fide pioneer transcription factor (Zaret & Mango, 2016), which likely recruits cellular machines for opening up repressed viral chromatin. It remains to be shown how BZLF1 can interact with nucleosomal DNA at high structural resolution, but future experiments should solve this conundrum.

# Materials and Methods

Additional Materials and Methods are available in the Materials and Methods section of the Supplementary Information. They include the following:

(i) Chromatin immunoprecipitation (ChIP) and sequential ChIPchromatin immunoprecipitation (ReChIP)
(ii) INO80 ChIP and qPCR
(iii) Generation of Raji p4816 cell lines with lentiviral shRNA vectors directed against INO80
(iv) Quantification of transcripts by qRT-PCR
(v) Next generation BZLF1 ChIP sequencing
(vi) Next generation CTCF ChIP sequencing
(vii) ATAC-seq analysis.

## Cells

Raji, THP-1, and HEK293 cells (Pulvertaft, 1964; Graham et al, 1977; Berges et al, 2005) were maintained in RPMI 1,640 medium (Thermo Fisher Scientific) supplemented with 10% FCS (Bio&Sell), 1% penicillin–streptomycin (Thermo Fisher Scientific), 1% sodium pyruvate (Thermo Fisher Scientific), and 100 nM sodium selenite (Merck) at 37°C and 5% $CO_2$. 293T cells were kept in DMEM (Thermo Fisher Scientific) including the supplements mentioned above. Raji p4816, p5693, and p5694 cells were kept under constant selection with 1 μg/ml puromycin. The HEK293-based cell line for the production of recombinant wild-type 2089 EBV stocks was cultivated in fully supplemented RPMI 1640 medium with 100 μg/ml hygromycin (Delecluse et al, 1998).

## Plasmids

Plasmids p4816, p5693, and p5694 (Fig S1) were constructed as described by Woellmer et al (2012). cDNAs coding for INO80 and SNF2h were cloned in frame with the eGFP gene and expressed from the CMV promoter in pEGFP-C1 and pEGFP-N3 (Clontech), respectively. All BZLF1 expression plasmids (aa 1–245, aa 1–236, aa 149–245, and aa 175–236) were expressed with a FLAG- and tandem Strep-tag (Gloeckner et al, 2007). The BZLF1 and gp110/BALF4 expression plasmids p509 and p2670 have been described (Hammerschmidt & Sugden, 1988; Neuhierl et al, 2002).

## Stable transfection and establishment of Raji cells

5 × 10⁶ Raji cells were suspended in 250 μl Opti-MEM I medium (Thermo Fisher Scientific), 5–10 μg plasmid DNA was added, and the cells were incubated on ice for 15 min. Electroporation (gene pulser II instrument; Bio-Rad) was performed in 4-mm cuvettes at 230 V and 975 μF. The cells were resuspended with 400 μl FCS, transferred to 5 ml fully supplemented medium as described above, and cultivated at 37°C for 2 d. For the establishment of single-cell clones, the cells were diluted in 96-well cluster plates and cultivated under selection for 4 wk. The medium was changed when necessary, and outgrowing cells were expanded. GFP expression was monitored by flow cytometry with a FACS Canto instrument by Becton Dickinson.

## Transient transfection of cell lines

Transfection of DNA into HEK293 cells using polyethylenimine (#24765; Polysciences) was done as described (Reed et al, 2006). For protein extracts, 2 × 10⁷ cells per 13-cm cell culture dish were seeded the day before transfection. Each plate was transfected with 30 μg plasmid DNA.

## ChIP and sequential ChIP (ReChIP)

All ChIP experiments were performed in triplicates as described previously (Woellmer et al, 2012) using anti-H3 (#1791; Abcam), anti-

H3K4me1 (#8895; Abcam), anti-BZLF1 (#17503; Santa Cruz) antibodies, or control IgG antibody (#PP64B; Millipore). All buffers were supplemented with the cOmplete protease inhibitor cocktail (Roche), and all steps were performed at 4°C if not noted otherwise. Details of the ChIP and ReChIP protocols can be found in the Materials and Methods section of the Supplementary Information.

Immunoprecipitated DNA was purified with the NucleoSpin Extract II kit (Macherey-Nagel) according to the manufacturer's protocol and eluted in 60 µl elution buffer. The samples were analyzed by qPCR with a LightCycler 480 (Roche) instrument.

### Quantitative real-time PCR (qPCR)

Immunoprecipitated DNA was quantified with a Roche LightCycler 480 instrument. PCR mixes consisted of template DNA (1 µl), primers (5 pmol each), and 2x SYBR Green I Master mix (5 µl) in a final volume of 10 µl. The PCR program for qPCR is listed in Table S1 in the Materials and Methods section of the Supplementary Information.

Primer design criteria were as follows: 62°C annealing temperature, primer efficiency of 2.0, and a single melting peak during Roche LightCycler 480 measurement. Primer synthesis was performed by Metabion (Munich), and sequences are listed in Table S2 in the Materials and Methods section of the Supplementary Information.

Absolute quantifications of the amount of DNA were calculated by comparing the crossing points of the unknown sample with a defined standard curve, which encompassed different dilutions of input DNA. The analysis was performed automatically with the LightCycler 480 software according to the "second derivative maximum method." Mean and SD were calculated from three independent biological replicates with one technical replicate each.

### In vitro DNA methylation

CpG methylation of plasmid DNA in vitro was done with the de novo methyltransferase M.SssI (New England Biolabs) and S-adenosyl methionine according to the manufacturer's recommendations.

### In vitro reconstitution of chromatin

*Drosophila* embryo histone octamers were prepared and used for in vitro nucleosome reconstitution via salt gradient dialysis according to Krietenstein et al (2012).

### EMSAs

EMSAs were performed according to Fried and Crothers (1981). Proteins were purified from HEK293 cells transiently transfected with a FLAG- and tandem Strep-tagged BZLF1 expression plasmid (Bergbauer et al, 2010). 2 d post transfection, the cells from six 13-cm cell culture dishes were pooled and lysed in 10 ml RIPA-buffer (50 mM Tris–HCl, pH 8.0, 150 mM NaCl, 1% Ipegal, 0.5% DOC, and 0.1% SDS). Lysates were sonicated and BZLF1 protein was purified using Strep-Tactin affinity chromatography according to the manufacturer's recommendations (IBA). All buffers were supplemented with the cOmplete protease inhibitor cocktail (Roche). Free and reconstituted DNA was radioactively labeled with [γ-32P] ATP by T4 polynucleotide kinase (NEB) according to the manufacturer's

recommendations. For each EMSA reaction, 0.4 nmol of radioactively labeled, free or reconstituted DNA was incubated with different concentrations of BZLF1 protein in the presence of 10 mM Tris–HCl, pH 7.6, 1 mM MgCl2, 60 mM KCl, 3 mg/ml BSA, 1% glycerol, 1% Ficoll, 1 mM DTT, 1 µg polydIdC (Roche), and 100 ng calf thymus DNA (Merck) in a total volume of 20 µl for 10 min at room temperature. Unbound template was separated from shifted complexes by polyacrylamide gel electrophoresis (5% (wt/vol) 29:1 acrylamide/ bisacrylamide, 0.5xTBE). Gels were analyzed with the aid of a radioisotope scanner (FLA 5100, Fuji), and quantitation of radioactivity signals was performed with the AIDA program (Raytest). For determination of the equilibrium dissociation constant (KD), these data were fitted to the Hill equation with one-site–specific binding using the PRISM 6 program (GraphPad).

### Co-IP

Co-IPs of GFP-fusion or Strep-tag fusion proteins were performed with GFP-Trap_A (ChromoTek) or Strep-Tactin affinity chromatography using Strep-Tactin Sepharose beads (IBA), respectively, according to the manufacturers' recommendations. All buffers contained the cOmplete protease inhibitor cocktail (Roche). Lysis buffers were supplemented with 5 U/µl Benzonase (Merck) and 0.5 µg/µl DNase I (Invitrogen) to exclude nucleic acid–mediated interactions.

### Western blotting

Proteins separated by SDS–PAGE were transferred (Mini Trans-Blot Cell; Bio-Rad) onto Hybond ECL membrane (GE Healthcare) at 100 V for 80 min and detected with the respective antibodies using the ECL reagent and X ray films (GE Healthcare). Primary antibodies were anti-BZ1 (kindly provided by Elisabeth Kremmer, Helmholtz Zentrum München), anti-SNF2h (#39543; Active Motif), anti-INO80 (#18810-1-AP; Proteintech), anti-CHD4 antibody (#ab70469; Abcam), anti-GFP (#290; Abcam), and anti-tubulin (#23948; Santa Cruz) and used at appropriate dilutions in blocking buffer (5 g skimmed milk powder in 100 ml PBS-T).

### siRNA knock-down

Transfections were performed with commercial siRNAs directed against transcripts of SNF2h/SMARCA5 (#E-011478-00; Dharmacon) or INO80 (#E-004176-00; Dharmacon) or with a random, non-targeting siRNA control pool (#D-001910-10-05; Dharmacon) (Fig S10C). siRNA transfections were done in serum-free Opti-MEM I medium (Thermo Fisher Scientific) in combination with HiPerFect transfection reagent (QIAGEN). 2089 EBV HEK293 cells (Delecluse et al, 1998) were transfected with the siRNAs three days before transient transfection of expression plasmids encoding BZLF1 and gp110/BALF4 to induce virus production in the siRNA-treated cells.

### Cell viability assay

siRNA transfected 2089 EBV HEK293 cells were seeded at three different densities into opaque-walled 96-well plate in 100 µl/well. 100 µl of CellTiter-Glo reagent (Promega) was added and mixed for 2 min on an orbital shaker to induce cell lysis. The plate was incubated at room temperature for 10 min to stabilize the luminescent signal,

which was subsequently recorded using a ClarioStar plate reader. Viability of the cells was calculated and expressed as percent of viable cells compared with cells treated with non-targeting siRNA.

### Generation of Raji p4816 cell lines with lentiviral shRNA vectors directed against INO80

Potentially suitable shRNAs were identified using the publicly available web tool siDirect2.0 (http://sidirect2.rnai.jp/). Based on the identified sequences, primers were designed and cloned into the pCDH lentiviral vector (System Biosciences), which was modified and termed p6573 as shown in Fig S10A. The lentiviral vector encodes the red fluorescent protein (DsRed) as a marker gene. Details of the production of lentiviral vectors can be found in the Materials and Methods section of the Supplementary Information. Raji p4816 cells stably transduced with shRNA_nt (non-targeting) or INO80-specific shRNAs were used in the experiments shown in Fig S11.

### Statistical analysis

We used Prism 7 (GraphPad) for statistical analysis, and the two-tailed ratio $t$ test was applied unless otherwise mentioned.

# Supplementary Information

# Acknowledgements

We thank Peter Becker, Munich, for the generous usage of his fly facilities and the supply of *Drosophila melanogaster* embryos for the preparation of histone octamers. We thank Bill Sugden, Madison, for critically reading our manuscript and valuable suggestions. We are grateful to Andreas Ladurner and Karl-Peter Hopfner, Munich, for providing us with the cloned human cDNAs of *SNF2h* and *INO80*, respectively. We also thank Dagmar Pich, Munich, for precious experimental advice. This work was financially supported by grants of the Deutsche Forschungsgemeinschaft (grant numbers SFB1064/TP A04, TP A11, and TP A13, SFB-TR36/TP A04), Deutsche Krebshilfe (grant numbers 107277 and 109661), and National Cancer Institute (grant number CA70723) and a personal grant to T Tagawa from Deutscher Akademischer Austauschdienst (Studienstipendien für ausländische Graduierte aller wissenschaftlichen Fächer).

## Author Contributions

M Schaeffner: data curation, formal analysis, investigation, methodology, and writing—original draft.
P Mrozek-Gorska: data curation, formal analysis, validation, investigation, methodology, and writing—original draft, review, and editing.
A Buschle: data curation, formal analysis, validation, investigation, visualization, methodology, and writing—review and editing.
A Woellmer: data curation, formal analysis, investigation, methodology, and writing—original draft.
T Tagawa: investigation and methodology.
FM Cernilogar: data curation, formal analysis, and methodology.

G Schotta: conceptualization, supervision, methodology, and writing—review and editing.
N Krietenstein: resources and methodology.
C Lieleg: resources and methodology.
P Korber: conceptualization, resources, supervision, methodology, and writing—original draft.
W Hammerschmidt: conceptualization, funding acquisition, validation, project administration, writing—original draft, review, and editing.

## Conflict of Interest Statement

The authors declare that they have no conflict of interest.

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
