## [Reviewer comments · Life Science Alliance]

Life Science Alliance

BZLF1 interacts with chromatin remodelers promoting escape from latent Epstein-Barr virus infection

Marisa Schaeffner, Paulina Mrozek-Gorska, Anne Woellmer, Takanobu Tagawa, Alexander Buschle, Filippo Cernilogar, Gunnar Schotta, Nils Krietenstein, Corinna Lieleg, Philipp Korber, and Wolfgang Hammerschmidt

Corresponding author(s): Wolfgang Hammerschmidt, Helmholtz Zentrum München, German Research Center for Environmental Health

Review Timeline:

Submission Date:	2018-06-15
Editorial Decision:	2018-07-25
Revision Received:	2019-02-07
Editorial Decision:	2019-03-11
Revision Received:	2019-03-17
Accepted:	2019-03-18

Scientific Editor: Andrea Leibfried

Transaction Report:

DOI: <https://doi.org/10.26508/lsa.201800108>

July 25, 2018

Re: Life Science Alliance manuscript #LSA-2018-00108-T

Wolfgang Hammerschmidt
GSF-National Research Center for Environment and Health
Department of Gene Vectors
Marchioninstr. 25
Munich 81377
Germany

Dear Dr. Hammerschmidt,

Thank you for submitting your manuscript entitled "BZLF1 interacts with the chromatin remodeler INO80 promoting escape from latent infections with Epstein-Barr virus" to Life Science Alliance. The manuscript was assessed by expert reviewers, whose comments are appended to this letter.

As you will see, the reviewer's opinion about your manuscript is slightly split. While reviewer #1 thinks that your work is interesting and could get published in Life Science Alliance if increasing the sample size and testing for significance of the results to better support your conclusions, reviewer #2 and #3 think that the specificity of the interaction between BLZF1 and INO80 and its relevance for INO80 recruitment to lytic genes remains to be shown. More direct evidence for your conclusions are needed and better controls need to be included.

We discussed your manuscript within our editorial team and with the reviewers and concluded that these concerns could in principle be addressed by following the constructive input offered by the reviewers. We would thus like to invite you to revise your work, addressing the concerns raised by all three reviewers. However, this would be a major revision and it remains unclear at this stage whether your conclusions will still stand upon revising your work. Therefore, please consider your options carefully. Importantly, we would need strong support on such a revised version from reviewer #3.

-- High-resolution figure, supplementary figure and video files uploaded as individual files: See our detailed guidelines for preparing your production-ready images, <http://life-science-alliance.org/authorguide>

B. MANUSCRIPT ORGANIZATION AND FORMATTING:

Full guidelines are available on our Instructions for Authors page, <http://life-science-alliance.org/authorguide>

Thank you for this interesting contribution to Life Science Alliance. We are looking forward to receiving your revised manuscript.

Sincerely,

Reviewer #1 (Comments to the Authors (Required)):

Review of Schaeffner et al

This paper questions the mechanism by which a viral transcription factor is able to activate gene expression to reactivate the virus from an epigenetically silenced genome in infected cells. The overall conclusion is that BZLF1 is a pioneer transcription factor that interacts with nucleosomes, it co-associates with two nucleosome remodelers and that one of these contributes to the reactivation of the virus from latency.

The evidence for the designation of BZLF1 as a pioneer transcription factor comes from the reduction in H3, the co-association of Zta at BZLF1 regulated genes, the transcription factor BZLF1 co-binding with histones in cells and the in vitro binding to nucleosomal DNA (Figs 1 -3).

The evidence for the association of BZLF1 with nucleosome remodelers is presented in the form of co-association within cells and further dissected using co-expression of mutant versions of the protein (Figs 5-6).

The relevance of INO80 to the re-activation of EBV is supported by activation of transcription of the viral genome and by changes in production of infectious virus. This is not so well supported. Specifically Figure 7 has no statistical support for the reductions in gene expression and figure 8 takes not account of the degree of 'knock down' of the remodelers into account. The general trend of the data is figure 7 is supportive but clearly more repeats are required to generate a larger data set for meaningful analysis.

Minor revisions: wherever histograms are shown the statistical relevance of the differences should be indicated and if they are not significant a larger data set should be analyzed.

Reviewer #2 (Comments to the Authors (Required)):

The paper investigates the activity of the EBV transactivator BZLF1 and provides evidence for the capacity of BZLF1 to act as a pioneer transcription factor that binds to nucleosomal DNA and interacts with chromatin remodelers to promote a chromatin conformation that is permissive for transcription. The paper outlines a set of interesting properties of BZLF1 is clearly written and the data are overall convincing. However, it is disappointing that, in comparing the activity of BZLF1 to that of the HSV VP16 transactivator that recruits chromatin remodelers to compact heterochromatin, the authors rely exclusively on indirect evidence, based co-immunoprecipitation and INO80 knockdown, rather than in vivo chromatin unfolding assays of the type described by Tumber et al. J.Cell.Biol. 145:1341, 1999. This type of assay would provide unequivocal answers to several questions that are not conclusively resolved by the current set of experiments including: i) is BZLF1 sufficient for chromatin remodeling in vivo; ii) is BZLF1 necessary and sufficient for the recruitment INO80 to heterochromatin; iii) which domain of BZLF1 recruits chromatin remodelers in

vivo.

Specific comments:

1. In Fig.1. the authors show that the expression of BZLF1 precedes by several hours the removal of H3 from early lytic promoters but has no effect of the occupancy of the promoters for latent and late genes. Are the differences in H3 occupancy significant? How does this observation relate to transcription of the viral genes? BZLF1 is for example an immediate early gene and should be transcriptionally active much before changes in H3 occupancy are detected in this type of assay. The dissociation between H3 occupancy and transcription is also suggested by the kinetics shown in Fig 7 where plateau level of the BMRF1 transcript are achieved at 8h post induction. The authors should comment on this.
2. Fig.7. the authors suggest that INO80 knockdown is associated with decrease transcription of early genes. I find this figure quite confusing, why was the transcription level of control transfected Raji p4816 at 15h set to 100%. Are the effects statistically significant?
3. Fig.8. The authors assess the effect of INO80 knockdown by measuring the release of infectious virus. Given that both the chromatin remodeling effect of BZLF1 (fig 1) and the effect of INO80 knockdown (fig 7) appears to be restricted to some early genes it is unclear why the authors choose this late readout, what happens to viral DNA synthesis?

Reviewer #3 (Comments to the Authors (Required)):

In the manuscript by Schaeffner et al., the authors examine activation of EBV lytic genes by the transcription factor BZLF1. Specifically the authors examine the role of BZLF1 in binding to nucleosomal DNA and recruiting chromatin remodelers to modulate lytic gene accessibility. Based on results presented in the manuscript, the authors propose that the BZLF1 is a pioneer transcription factor that binds target sequences bound by nucleosomes and recruits the INO80 remodeler to fully activate gene expression.

The model is provocative and interesting, and would place a viral transcription factor in the unique field of other pioneer transcription factors that are important for differentiation and cellular reprogramming. However, I believe the experimental design and results limit the ability of the authors to confidently support their experimental interpretation and overall conclusions. For example, the authors utilize a modified histone ChIP as a proxy for nucleosome occupancy. Technical limitations are mentioned, however, if the appropriate experiment cannot be performed then conclusions should be adjusted accordingly. The authors also extensively examine BZLF1 binding to a diverse set of nucleosomal substrates and make assertions as to the ability of BZLF1 to bind relatively inaccessible recognition sequences bound to nucleosomes, however, accessibility (e.g enzymatic accessibility) is not systematically investigated. Additionally, the authors do not convincingly demonstrate that BZLF1 can recruit INO80 to lytic genes. These limitations significantly reduce enthusiasm for the findings presented in the manuscript.

Below are my specific comments:

- For the inducible system described on p.12, the authors refer to unpublished data demonstrating that expression of BZLF1 is "in a range" found in cells undergoing the EBV lytic cycle. The authors should show this work in order to support use of their system.
- In Fig 1, the authors utilize sonicated chromatin to map nucleosomes on early lytic genes. Typically, micrococcal nuclease is used to fragment chromatin for nucleosome mapping experiments. If the authors choose another method, they should validate extensively using a locus with known nucleosome positions and include gels showing sonicated fragment sizes.

- For experiments in Fig. 2 that aim to demonstrate that BZLF1 binds directly to nucleosomes, the use of H3K4me1 is not an appropriate proxy for total histones. Although Re-ChIP experiments are more difficult than standard ChIP, the authors are limited in their conclusions using a modified histone antibody. Results observed may be limited to change in histone modification only, and not a change in nucleosome occupancy.
- Importantly, standard deviation/error is not shown in Fig. 2, thus significance of results are not interpretable.
- By eye, it is not quite clear that the results of the EMSAs in Fig 4B support the authors conclusions. The EMSAs have an appearance that the 3+4 binding may be additive of 0+4 and 0+3 fragments. The authors should describe in more detail how the quantifications were performed.
- In Fig. 4D the authors claim that rotational position does not affect binding, but the actual results suggest otherwise.
- Overall, the authors spend a lot of time discussing differential binding of BZLF1 to different nucleosomal substrates. However, accessibility should be confirmed using DNase or ATAC assays.
- Fig 5 lacks proper negative controls, such as beads alone.
- On p. 17, the authors state "The BZLF1 expression plasmids were adjusted to obtain similar protein levels (Fig. 6B)." What does "adjusted" mean?
- The authors claim that results in Fig. 6C suggest aa175-236 of BZLF1 are responsible for SNF2h interaction. However, other constructs that contain the same domain do not bind. Can the authors explain why?
- The authors need to provide a reference for the W653Q INO80 point mutant or characterize themselves. Also, INO80 has been successfully ChIPed in several systems, so the use of this mutant may not be necessary.
- The point regarding the recruitment of INO80 to lytic genes by BZLF1 is a major point of the manuscript, yet not well addressed. Fig S8 shows mutant INO80 is increased at viral promoters with induction of BZLF1, yet INO80 is also increased on controls. The statistical significance between controls and viral genes is not assessed. In addition, there is detectable binding of INO80 prior to BZLF1 expression, thus a majority of INO80 binding does not need BZLF1, and contradicts the authors conclusions. (A KO of BZLF1 should be used as an additional control)
- In Fig. 7 the authors demonstrate that a KD of INO80 impairs viral gene expression. It is well known that INO80 regulates a vast array of genes in different cell types. The authors should include more controls to demonstrate the effect is specific to viral genes or at least qualify their conclusions to note possible indirect effects.

Minor points:

- On p.4 the authors write "This study did not determine whether...". Are they referring to the referenced paper in the previous sentence?
- On p.13 the authors refer to H3K4me1 ChIP as "to be published and Fig. 2A". Is "to be published" necessary if already shown in Fig. 2A?
- Fig 5 and 6 could be combined.
- The Discussion is quite lengthy and could be abbreviated.

We have read the comments and concerns raised by the three reviewers very carefully. It appears to us that two topics dominate the discussion:

(i) The functional link between BZLF1 and INO80, in particular the recruitment of INO80 to lytic genes by BZLF1 is not directly shown.

(ii) Chromatin opening induced by BZLF1 binding is uncertain and needs more support.

We would like to describe our additional experiments that address these two topics in our revised manuscript.

Ad (i) Functional link between BZLF1 and INO80

We are convinced that BZLF1 recruits cellular chromatin remodelers, in particular INO80, to epigenetically repressed viral DNA. In principle, chromatin immunoprecipitation with an antibody directed against INO80 components is the appropriate experiment that should deliver the most direct evidence. In our original submission we invested a lot of effort to make this experiment work and showed the results with transiently transfected 2089 EBV HEK293 cells in the (previous) Fig. S8. The results in this figure supported our initial working hypothesis, but the differences between non-induced and BZLF1 induced cell chromatin were small and not entirely convincing. We now revisited this experimental problem and turned to our doxycycline-inducible Raji cell model.

We compared chromatin from non-induced cells and from cells induced for 6 and 15 hours. For the new set of ChIP experiments, we employed and improved a published protocol (*Zhou et al [2016] INO80 governs superenhancer-mediated oncogenic transcription and tumor growth in melanoma. Genes Dev 30:1440–1453. doi: 10.1101/gad.277178.115*), tested three antibodies in total and chose an INO80 specific antibody that we had not used in the first and previous experiments (former Fig. S8). We selected chromosomal loci as control and reference, which BZLF1 do not bind in ChIP-seq experiments and which are closed and not accessible in ATAC-seq experiments (see below for the details of the ATAC-seq approach).

In four independent ChIP experiments we obtained comparable results indicating that INO80 can be found at sites in viral chromatin where BZLF1 binds. On the contrary, INO80 is not enriched at cellular sites that BZLF1 does not bind and hence served as our controls. The differences between the non-induced and induced states appear obvious although they are relatively small. We now show these ChIP data with INO80 in the newly added Figure 6 in our revised manuscript replacing the previous Fig. S8.

We would like to point out that ChIPs with chromatin remodelers are notoriously difficult and controversially discussed. Reviewer #3 stated that “... *INO80 has been successfully CHIPed in several systems.*” and we are familiar with the publication mentioned above (*Zhou et al [2016] INO80 governs superenhancer-mediated oncogenic transcription and tumor growth in melanoma. Genes Dev 30:1440–1453. doi: 10.1101/gad.277178.115*). Following this published protocol, we found a moderate recruitment of INO80 at lytic viral promoters upon BZLF1 expression as shown in the new Figure 6 of our revised manuscript. Our findings are not so surprising given a wealth of published literature with negative outcome. For example, in flies, ChIP with the ACF and RSF remodelers led to false-positive

peaks, only (Jain et al [2015] *Active promoters give rise to false positive 'Phantom Peaks' in ChIP-seq experiments. Nucleic Acids Res 43:6959–6968. doi: 10.1093/nar/gkv637*). Similarly in ChIP experiment in yeast, the lab of Steve Henikoff did not find Isw1/2 and Chd1 remodeler at their sites of action (Zentner et al [2013] *ISWI and CHD chromatin remodelers bind promoters but act in gene bodies. PLoS Genet 9:e1003317. doi: 10.1371/journal.pgen.1003317*). Again in ChIP experiments, the Pugh lab found only subsets of genes to be bound by different remodelers tested and no genes with the Chd1 remodeler (Yen et al [2012] *Genome-wide nucleosome specificity and directionality of chromatin remodelers. Cell 149:1461–1473. doi: 10.1016/j.cell.2012.04.036*).

It seems to us that the field is controversial. Given our INO80 ChIP data together with circumstantial evidence (protein co-IPs and INO80 knock-down experiments with two different read-outs) we are convinced that INO80 is the key remodeler that is recruited by the transcriptional activation domain of BZLF1 to open up repressed lytic viral promoters.

To respond to the reviewers' critique, we tamed down our argumentation in the revised manuscript and also altered its title by omitting INO80. We do hope that these measures improve our manuscript and make it less conflicting.

Ad (ii) Chromatin opening induced by BZLF1 binding

Our previous experiments did not provide a clear functional linkage between BZLF1 binding to silent chromatin and its de-repression and reorganization by cellular remodelers. As suggested by the reviewers, we performed additional experiments to address this issue. First, we did ChIP-seq experiments with the BZLF1-specific BZ1 antibody followed by ATAC-seq experiments using identical conditions in two Raji cell lines. As an internal control, we also show ChIP-seq results with a CTCF-specific antibody in the revised manuscript.

As described in our work, one Raji cell line encodes a full-length BZLF1 protein, the other a BZLF1 mutant that is devoid of BZLF1's transcriptional activation domain (AD-truncated BZLF1). Both alleles are inducibly expressed upon addition of doxycycline. Especially in our ATAC-seq experiments great care was taken to (i) analyze viable and physically intact cells, only, and (ii) to restrict our analysis to cells that supported the conditional expression of BZLF1. We used physical sorting of the cells according to forward and sideward FACS criteria and sorted GFP-positive cells upon addition of doxycycline, because both GFP and BZLF1 are conditional, co-regulated genes (Fig. S1).

The new Figure S8 provides an overview of our bioinformatic workflow, which is described in detail in the section Materials and Methods. We postulated that BZLF1 binding should induce the local opening of silent viral chromatin, because our previous experiments in this manuscript as well as in our published paper (Woellmer et al [2012], *PLoS Pathog 8:e1002902. doi: 10.1371/journal.ppat.1002902; Fig. 2*) suggested that BZLF1 recruits cellular chromatin remodelers that mobilize and/or evict nucleosomes at BZLF1 binding sites. The results from the ATAC-seq experiments nicely support our working hypothesis as described in our revised manuscript in detail. The newly added Figures 7 and 8 together with the Supplemental Figure S9 document this fact in viral chromatin.

Our experiments also allowed studying the binding of BZLF1 in Raji cell chromatin and observing the ensuing consequences. Using the peak caller MACS2, we found more than 10^5 functional BZLF1 binding sites in cellular chromatin. Next, we analyzed the average coverage of ATAC-seq reads at these sites in the two Raji cell lines in their non-induced and induced states. The visualization in Figure A below clearly documents the opening of silent chromatin at cellular BZLF1 binding sites that occurs at induced high levels of BZLF1, only. More specifically, the average peak of chromatin opening co-locates exactly with the peak center of the $>10^5$ cellular BZLF1 binding sites in induced cells that express full-length BZLF1. A truncated BZLF1 protein without its transcriptional activation domain does not induce chromatin remodeling (Fig. A, right panel), although it binds silent chromatin as efficiently as full-length BZLF1. Together, these findings clearly support our view that BZLF1's activation domain is critically involved in recruiting chromatin remodelers such as INO80 to silent chromatin. As a consequence, previously closed chromatin becomes readily accessible.

The cellular chromatin data in the right panel of Figure A below seem to look “sharper” or “more pronounced” than data obtained with viral chromatin shown in Figure 8A of our revised manuscript. In cellular chromatin many of the $>10^5$ individual BZLF1 binding sites are isolated and widely distributed in contrast to EBV chromatin in which the 66 identified BZLF1 ChIP-seq peaks (with a total of 85 BZLF1 binding motifs) are often narrowly arranged in clusters. As a result, neighboring BZLF1 sites in viral chromatin result in broad regions with open chromatin (new Fig. 7 of our revised manuscript) that often do not show a peak-like shape compared with ATAC-seq reads in cellular chromatin.

We discuss our ATAC-seq findings at length in this rebuttal letter to provide unequivocal evidence for our hypothesis. The very many BZLF1 binding sites in cellular DNA and their locus-specific chromatin opening upon BZLF1 binding support our findings with EBV chromatin very nicely.

The ATAC-seq and BZLF1 ChIP-seq data with cellular chromatin will be presented in a separate manuscript that is in its advanced state of writing. In it we show that the induced expression of full-length BZLF1 causes an almost complete disruption of the cellular 3D chromatin architecture, wide-spread loss of chromatin-chromatin interactions concomitant with a general loss of chromatin accessibility, and a massive downregulation of cellular transcripts. Only at the many BZLF1 binding sites cellular chromatin opens up and shows an impressive but locally restricted increase in ATAC-seq reads as summarized in the Figure A below. We interpret these findings to mean that BZLF1 directly and indirectly inactivates cellular genes to redirect the cellular transcriptional machinery to viral DNA supporting EBV's very efficient transcriptional activation during the lytic phase.

Figure A. High BZLF1 levels induce open cellular chromatin at thousands of cellular BZLF1 binding sites

A. In ChIP-seq experiments with a BZLF1 specific antibody, the peak caller MACS2 identified 145,544 BZLF1 peaks in Raji p4816 cells 15 hours after induction with doxycycline. The metaplot visualizes the average peak coverage of these BZLF1 sites in cellular chromatin 15 hours after induction of full-length BZLF1. Mapped sequencing reads of chromatin prepared from the induced cells prior to immunoprecipitation are shown as a reference (input).

B. The metaplot summarizes the chromatin accessibility at the 145,544 BZLF1 binding sites prior to and after induction of full-length or AD-truncated BZLF1 in Raji cell chromatin. The average ATAC-seq coverages in the four different Raji cell samples are plotted according to the nucleotide coordinates of the 145,544 BZLF1 peaks identified in panel A. In non-induced Raji p4618 cells (BZLF1 full-length, non-induced) the average ATAC-seq coverage is congruent with the coverage found in induced and non-induced Raji cells that carry the conditional AD-truncated BZLF1 allele. At induced BZLF1 levels (full-length, induced) the average ATAC-seq coverage resembles BZLF1's average peak coverage shown in panel A. The inset figure provides the ATAC-seq coverage of 1,455,500 randomly sampled sequences in chromatin of Raji p4816 cells (full-length BZLF1) and AD-truncated BZLF1 expressing Raji cells prior to and 15 hours after doxycycline-mediated induction.

We believe that these new findings open a new chapter in understanding the regulation of EBV's lytic phase. Therefore, we would like to publish the ATAC-seq data covering cellular aspects separately from this revised manuscript and in a different functional context.

Our comments to the specific points of the three reviewers

Reviewer #1

The relevance of INO80 to the re-activation of EBV is supported by activation of transcription of the viral genome and by changes in production of infectious virus. This is not so well supported. Specifically Figure 7 has no statistical support for the reductions in gene expression and figure 8 takes not account of the degree of 'knock down' of the remodelers into account. The general trend of the data is figure 7 is supportive but clearly more repeats are required to generate a larger data set for meaningful analysis.

In the revision of our manuscript we focused on the physical interaction of INO80 and BZLF1 in ChIP experiments and the opening of viral chromatin at BZLF1 sites in ChIP-seq experiments. As pointed out in the introduction to this rebuttal letter, the newly added data provide additional evidence complementing and supporting the findings presented in the (previous) Figure 8. (The former Figure 8 is now Figure 9 in our revised manuscript). We determined the statistical significance in the knock-down experiments shown in the previous Figure 7 (now Figure S11 in the revised manuscript) and found that some hold up to accepted p-values ($p < 0.05$).

Minor revisions: wherever histograms are shown the statistical relevance of the differences should be indicated and if they are not significant a larger data set should be analyzed.

We followed this reviewer's recommendations and provide p-values in the revised Figures 9 and S11 to support the observed but often small differences.

Reviewer #2

1. In Fig.1. the authors show that the expression of BZLF1 precedes by several hours the removal of H3 from early lytic promoters but has no effect of the occupancy of the promoters for latent and late genes. Are the differences in H3 occupancy significant? How does this observation relate to transcription of the viral genes? BZLF1 is for example and immediate early gene and should be transcriptionally active much before changes in H3 occupancy are detected in this type of assay. The dissociation between H3 occupancy and transcription is also suggested by the kinetics shown in Fig 7 where plateau level of the BMRF1 transcript are achieved at 8h post induction. The authors should comment on this.

This reviewer is correct in stating that there is an apparent discrepancy between the detectable loss of H3 at lytic promoters (Fig. 1B) and the rapid onset of transcription of certain lytic genes (Fig. S11 [former Fig. 7]). In our Raji cells we find about 15 to 20 genomic copies of EBV DNA. Presumably they are all epigenetically repressed during latency but it is uncertain if some or all DNA copies become activated in a synchronous manner upon induction of EBV's lytic phase. We think that the binding of BZLF1 precedes a detectable decrease in nucleosomal occupancy at early lytic promoters by hours because a substantial fraction of nucleosomes must become removed before this nucleosomal loss becomes detectable. We are also uncertain whether nucleosomes must be evicted prior to the onset of lytic transcription or whether it suffices that chromatin remodelers mobilize nucleosomes

and cause a de-compaction of the silent and repressed latent viral chromatin to induce lytic viral transcription.

2. Fig.7. the authors suggest that INO80 knockdown is associated with decrease transcription of early genes. I find this figure quite confusing, why was the transcription level of control transfected Raji p4816 at 15h set to 100%. Are the effects statistically significant?

We chose to use this visualization because compared with the non-induced situation the RT-qPCR levels are much elevated 15 h hours post induction and thus allow a more reliable quantification at initial, very low and high mRNA levels at the endpoint. Certain effects appear statistically significant or show a strong tendency. This information is now provided in the modified Figure S11. (We renumbered the former Fig. 7, which is Fig. S11 in the revised manuscript).

3. Fig.8. The authors assess the effect of INO80 knockdown by measuring the release of infectious virus. Give that both the chromatin remodeling effect of BZLF1 (fig 1) and the effect of INOS80 knockdown (fig 7) appears to be restricted some early genes it is a unclear why the author choose this late readout, what happens to viral DNA synthesis?

We chose this 'late' readout because it summarizes the function of INO80 during EBV's lytic phase. If this read-out had not worked we would be concerned if we studied a relatively unimportant aspect of EBV's strategy. It is clear from our lentiviral knock-down approach that the knock-down is quite modest (Fig. 9 in the revised manuscript), thus the reduction in virus synthesis is not very strong. One should also consider that INO80 belongs to an abundant protein complex that is one out of several chromatin remodelers with often redundant functions. We investigated only two chromatin remodelers in the manuscript, INO80 and SNF2a and excluded CDH4, but BZLF1 seems to be rather promiscuous in recruiting additional cellular (and viral) components to viral DNA.

Reviewer #3

- For the inducible system described on p.12, the authors refer to unpublished data demonstrating that expression of BZLF1 is "in a range" found in cells undergoing the EBV lytic cycle. The authors should show this work in order to support use of their system.

We invested considerable work to assess the number of BZLF1 protein dimers in cells that spontaneously support EBV's lytic phase. We took this number as a proxy and found that in our model, the doxycycline-induced Raji cell line, BZLF1 levels are higher by a factor of about 3 to 4 reaching approximately 7×10^6 dimers per cell upon induction. This finding will be presented independently in a separate manuscript that is in its final stages of writing and will be submitted in February or March at the latest. Once submitted we will refer to it in the current manuscript.

- In Fig 1, the authors utilize sonicated chromatin to map nucleosomes on early lytic genes. Typically, micrococcal nuclease is used to fragment chromatin for nucleosome mapping experiments. If the authors choose another method, they should validate extensively using a locus with known nucleosome positions and include gels showing sonicated fragment sizes.

Our description of chromatin preparation was incorrect. Chromatin was fragmented by both sonication plus MNase treatment to an average size of lower than 200 bps, which was validated by agarose gel electrophoresis. We corrected this information in the revised manuscript.

- For experiments in Fig. 2 that aim to demonstrate that BZLF1 binds directly to nucleosomes, the use of H3K4me1 is not an appropriate proxy for total histones. Although Re-ChIP experiments are more difficult than standard ChIP, the authors are limited in their conclusions using a modified histone antibody. Results observed may be limited to change in histone modification only, and not a change in nucleosome occupancy.

The primary intention of this experiment was to demonstrate that BZLF1 binds to nucleosomal DNA in cellular chromatin *in vivo*. We tested several pan histone antibodies but were confronted with either a low recovery of chromatin or high background. We therefore chose this excellent antibody directed against the histone mark H3K4me1. It is obvious that our approach reveals the interaction of BZLF1 with nucleosomes containing H3K4me1 marks, only. The results of this experiment show what we hoped to find in principal and support our hypothesis nicely. In our opinion, it is fair to conclude that BZLF1 will also contact nucleosomes with other histone marks, because it is very unlikely that BZLF1 has properties of chromatin 'readers' that recognize histone marks specifically.

- Importantly, standard deviation/error is not shown in Fig. 2, thus significance of results are not interpretable.

This critique is correct. With ReChIP experiments it is very difficult to achieve similar values in biological replicates, mainly because the rate of recovery is very low and thus variable from one experiment to the next. The experiment shown in Fig. 2 is one out of three independent replicates, which are all provided for additional inspection in Fig. S2. The results in this supplemental figure clearly support our interpretation of Fig. 2.

- By eye, it is not quite clear that the results of the EMSAs in Fig4B support the authors conclusions. The EMSAs have an appearance that the 3+4 binding may be additive of 0+4 and 0+3 fragments. The authors should describe in more detail how the quantifications were performed.

We exposed the dried EMSA gels to appropriate screens that were scanned with the aid of a phosphorimager. The ratio of the signal intensities of shifted versus non-shifted bands were calculated to obtain the Hill equations in Fig. 4C, for example. The K_d values are based on at least triplicates and only examples of such gels are shown. This approach is described in the Materials and Methods section. The conclusions drawn are based on the calculated ratios.

- In Fig. 4D the authors claim that rotational position does not affect binding, but the actual results suggest otherwise.

We reconsidered our data in Fig. 4D and agree with this reviewer's opinion now. It is further supported by a structural alignment of the BZLF1 DNA-binding domain with the nucleosome by Carlo Petosa, Geneva. Figure B below shows half of the sterically accessible sites (for nucleosome base pairs 0 to +90; equivalent sites occur for base pairs -90 to 0). There is a

continuous cluster of 4 to 5 sites around nucleosomal exit/entry, whereas sites are more sparsely distributed around the rest of the nucleosomal core, separated by 10 bp steps. Interestingly, at nearly all of these sites, BZLF1's DNA binding domain residues are very close to core histones residues, suggesting that BZLF1 directly interacts with the histone core globular domains and also very likely with the nearby histone N-terminal tails. This notion is supported by our additional experiments that document a stable association of BZLF1 with single histones (data not shown).

Figure B. Model of a structural alignment of the BZLF1 DNA binding domain (DBD) with the nucleosome.

The model indicates the discrete sites where BZLF1 might probably bind without requiring major distortions of either binding partner. At all other positions there is a large steric clash between BZLF1 and the nucleosome. Note that this analysis does not account for the N-terminal domain that precedes the BZLF1 DBD, which would add further steric constraints (data and figure by Carlo Petosa, Geneva, personal communication).

- Overall, the authors spend a lot of time discussing differential binding of BZLF1 to different nucleosomal substrates. However, accessibility should be confirmed using DNase or ATAC assays.

We now provide the requested information in the new Figures 7, 8, S8, and S9. We discuss the rationale of the ChIP-seq experiments with antibodies directed against CTCF and BZLF1 and the ATAC-seq approach in the beginning of this rebuttal letter together with Figure A.

- On p. 17, the authors state "The BZLF1 expression plasmids were adjusted to obtain similar protein levels (Fig. 6B)." What does "adjusted" mean?

In transiently transfected cells the expression levels of plasmids encoding different versions of truncated BZLF1 differed. To compensate for this effect, we used adjusted DNA amounts of the expression plasmids for transient transfection to reach comparable levels of protein expression for the co-IPs.

- The authors claim that results in Fig. 6C suggest aa175-236 of BZLF1 are responsible for SNF2h interaction. However, other constructs that contain the same domain do not bind. Can the authors explain why?

We are not certain about this point of critique. Figure 6C, which is now Figure 5D shows that a robust interaction of aa175 to 236 of BZLF1 with SNF2h exists. For unclear reasons, the interaction of aa175 to 245 appears weaker in this experiment, which was not consistently seen in others. The interaction of SNF2h appears definitely weaker with aa149 to 245 of BZLF1. The very last lane does not contain GFP-SNF2h as a control.

- The authors need to provide a reference for the W653Q INO80 point mutant or characterize themselves. Also, INO80 has been successfully ChIPed in several systems, so the use of this mutant may not be necessary.

The reference was mentioned in the beginning of this rebuttal letter and it was also cited in the manuscript: *Gelbart et al (2005) Genome-wide identification of Isw2 chromatin-remodeling targets by localization of a catalytically inactive mutant. Genes Dev 19:942–954. doi: 10.1101/gad.1298905*. As discussed above in the introduction to this rebuttal letter, we removed this experiment and replaced it with the more supportive Figure 6 in the revised manuscript.

- The point regarding the recruitment of INO80 to lytic genes by BZLF1 is a major point of the manuscript, yet not well addressed. Fig S8 shows mutant INO80 is increased at viral promoters with induction of BZLF1, yet INO80 is also increased on controls. The statistical significance between controls and viral genes is not assessed. In addition, there is detectable binding of INO80 prior to BZLF1 expression, thus a majority of INO80 binding does not need BZLF1, and contradicts the authors conclusions. (A KO of BZLF1 should be used as an additional control)

As pointed out already, we replaced this experiment with a more direct approach, which is shown in the new Figure 6.

- In Fig. 7 the authors demonstrate that a KD of INO80 impairs viral gene expression. It is well known that INO80 regulates a vast array of genes in different cell types. The authors should include more controls to demonstrate the effect is specific to viral genes or at least qualify their conclusions to note possible indirect effects.

The former Fig. 7 is Fig. S11 in the revised manuscript. We added this reviewer's concern ("qualify their conclusions to note possible indirect effects; INO80 regulates a vast array of genes") as requested to the text of the Result section of the revised manuscript.

Minor points:

- On p.4 the authors write "This study did not determine whether...". Are they referring to the referenced paper in the previous sentence?

Yes. We changed the text to make the link clear.

- On p.13 the authors refer to H3K4me1 ChIP as "to be published and Fig. 2A". Is "to be published" necessary if already shown in Fig. 2A?

We eliminated "to be published" in the revised manuscript.

- Fig 5 and 6 could be combined.

Done as requested. Looks better, indeed.

- The Discussion is quite lengthy and could be abbreviated.

We shortened the Discussion section where we found it appropriate.

March 11, 2019

RE: Life Science Alliance Manuscript #LSA-2018-00108-TR

Prof. Wolfgang Hammerschmidt
Helmholtz Zentrum München, German Research Center for Environmental Health
Research Unit Gene Vectors
Marchioninstr. 25
Munich 81377
Germany

Dear Dr. Hammerschmidt,

Thank you for submitting your revised manuscript entitled "BZLF1 interacts with chromatin remodelers promoting escape from latent Epstein-Barr virus infection". As you will see, reviewer #3 appreciates the introduced changes, while reviewer #2 thinks that your conclusions are still not sufficiently supported, requiring further text changes to tone them down. We would thus like to invite you to provide a final version of your manuscript, addressing reviewer #2's concerns. Please also note the following editorial requests:

- the figure legends are missing for Fig 7B and 8D, please add
- callouts are missing for Fig S3E and S8B, please add
- please list the databases and accession codes for your NGS data, maybe this can be done by adding columns to Table S3

A. FINAL FILES:

B. MANUSCRIPT ORGANIZATION AND FORMATTING:

Sincerely,

Reviewer #2 (Comments to the Authors (Required)):

Cooper and coauthors have submitted an expensively revised version of their manuscript that, in my opinion, strengthen some of the conclusions but still fails to provide conclusive evidence for the main tenet of the paper namely the involvement of chromatin remodelers in the BPLF1 mediated reactivation of EBV and escape from latency. In particular:

1. Fig.1 and Fig.2 convincingly shows that expression of the BZLF1 transactivator precedes by several hours the loss of histones at some early lytic promoters, and that the recruitment of BZLF1 to the promoters is accompanied by acquisition of the H3K4me1 activation marker
2. In Fig.3 and Fig.4 the authors perform a careful analysis that conclusively demonstrates the capacity of BZLF1 to interact with chromatin
3. Fig 5. documents the capacity of BPLF1 to interact with chromatin remodelers with particular focus on the interaction of BZLF1-TAD with INO80
4. Fig 6 documents the recruitment of INO80 to some but not all lytic promoters. The recruitment is a late event that precedes by several hours the initiation of transcription. The authors argue that transcription may initiate asynchronously in the resident genomes and the discrepancy is therefore explained by the different sensitivity of the assays. This may very well be the case but the explanation does not account for the early detection of the H3K4me1 marker
5. In Fig 7 and Fig 8 the authors provide an interesting new set of experiments that nicely confirm the correlation between the recruitment of BZLF1 multiple sites on the viral chromatin and increased accessibility of the sites measured by ATAC-seq. The opening of chromatin appears to be dependent on the BZLF1 activation domain that binds to INO80 but there is no direct evidence for the involvement of the remodeler in chromatin opening.
6. In Fig 9 partial knockdown of INO80 is shown to correlate with a significant decrease of virus production. This experiment is quite disconnected to the previous set of experiments and does not address the role the remodeler in the opening up of chromatin that accompanies virus reactivation. The authors argue that by measuring virus production they wish to assess a biological relevant effect. This is possibly correct but I would argue that the knockdown will also affect a variety of cellular genes that could indirectly influence the efficiency of virus production, maturation and release. Data on the role of INO80 in the regulation of transcription are shown in Fig S11B. The knockdown affects the transcription of only 2 out of 4 genes tested and comparison with the CHIP data shown in Fig 6 suggests a very poor correlation between the effect of knockdown and the recruitment of INO80 to the promoters. For example: there is significant recruitment of INO80 the BRLF1 and BBLF4 promoter (fig 6) but INO80 knockdown does not affect the transcription of these genes (Fig S11B), conversely, the transcription of BMRF1 and BNLF2a is decreased upon INO80 knockdown (Fig S11B) but INO80 is not recruited to these promoters. Since the knockdown seems to work quite well the authors should test whether the knockdown affects the opening of viral chromatin detected the ATAC-seq assay.

In conclusion, the authors have made a careful revision of the manuscript and added new data but they still fail to conclusively demonstrate that the opening of chromatin induced by the BZLF1 transactivator is mediated by the recruitment of chromatin remodelers. In the absence of this conclusive evidence the authors should tune down some of the claims made in the discussion

Reviewer #3 (Comments to the Authors (Required)):

The authors have performed substantial revisions to the manuscript and have satisfied all my

previous concerns.

March 18, 2019

RE: Life Science Alliance Manuscript #LSA-2018-00108-TRR

Prof. Wolfgang Hammerschmidt
Helmholtz Zentrum München, German Research Center for Environmental Health
Research Unit Gene Vectors
Marchioninstr. 25
Munich 81377
Germany

Dear Dr. Hammerschmidt,

Thank you for submitting your Research Article entitled "BZLF1 interacts with chromatin remodelers promoting escape from latent Epstein-Barr virus infection". We appreciate the introduced changes and it is a pleasure to let you know that your manuscript is now accepted for publication in Life Science Alliance. Congratulations on this interesting work.

DISTRIBUTION OF MATERIALS:

Again, congratulations on a very nice paper. I hope you found the review process to be constructive and are pleased with how the manuscript was handled editorially. We look forward to future exciting submissions from your lab.

Sincerely,

Andrea Leibfried, PhD
Executive Editor
Life Science Alliance
Meyerohofstr. 1
69117 Heidelberg, Germany
t +49 6221 8891 502
e a.leibfried@life-science-alliance.org
www.life-science-alliance.org